# Lipid packing and cholesterol content regulate membrane wetting and remodeling by biomolecular condensates

Agustín Mangiarotti [1,5] ✉, Elias Sabri[1,6], Kita Valerie Schmidt[1,2,6], Christian Hoffmann [3], Dragomir Milovanovic [3,4], Reinhard Lipowsky[1] & Rumiana Dimova [1] ✉

Biomolecular condensates play a central role in cellular processes by interacting with membranes driving wetting transitions and inducing mutual remodeling. While condensates are known to locally alter membrane properties such as lipid packing and hydration, it remains unclear how membrane composition and phase state in turn affect condensate affinity. Here, we show that it is not only the membrane phase itself, but rather the degree of lipid packing that determines the condensate affinity for membranes. Increasing lipid chain length, saturation, or cholesterol content, enhances lipid packing, thereby decreasing condensate interaction. This regulatory mechanism is consistent across various condensate-membrane systems, highlighting the critical role of the membrane interface. In addition, protein adsorption promotes extensive membrane remodeling, including the formation of tubes and double-membrane sheets. Our findings reveal a mechanism by which membrane composition fine-tunes condensate wetting, highlighting its potential impact on cellular functions and organelle interactions.

The view on intracellular organization expanded with the discovery that in addition to membrane-bound organelles, there are organelles lacking a surrounding membrane, also known as biomolecular condensates. These membrane-less organelles exhibit liquid-like properties and provide additional means of compartmentation, playing key roles in cell physiology and disease[1,2]. In recent years, several cellular processes involving the interactions between membranes and condensates have been described, such as the biogenesis and fission of protein-rich granules in the endoplasmic reticulum[3,4], receptor clustering and signaling in T-cells[5,6], the assembly of endocytic vesicles[7], and the interaction between stress granules and lysosomes[8,9].

The contact between membrane-bound and membraneless organelles not only regulates condensate dynamics and assembly[3,4,6], but also promotes their mutual remodeling[10–16], and the transmembrane coupling of phase separated proteins[17]. Studying the mechanism behind such interactions in cells is a challenging endeavor due to the dynamic nature and the small size of condensates, often below optical resolution[14]. In this context, biomimetic systems have been instrumental in overcoming these difficulties, revealing general mechanisms underlying the membrane-condensate interactions. At the microscale, wetting transitions govern the interaction between membranes and non-anchored three-dimensional (3D) condensates[18], and can be modulated by different parameters, such as ionic concentration or

[1]Max Planck Institute of Colloids and Interfaces, Science Park Golm, 14476 Potsdam, Germany. [2]Institute of Biochemistry, Freie Universität Berlin, Thielallee 63, 14195 Berlin, Germany. [3]Laboratory of Molecular Neuroscience, German Center for Neurodegenerative Diseases (DZNE), 10117 Berlin, Germany. [4]Einstein Center for Neuroscience, Charité-Universitätsmedizin Berlin, Corporate Member of Freie Universität Berlin, Humboldt-Universität Berlin, and Berlin Institute of Health, 10117 Berlin, Germany. [5]Present address: Laboratory of Molecular Neuroscience, German Center for Neurodegenerative Diseases (DZNE), 10117 Berlin, Germany. [6]These authors contributed equally: Elias Sabri, Kita Valerie Schmidt. ✉e-mail: Agustin.Mangiarotti@mpikg.mpg.de; Rumiana.Dimova@mpikg.mpg.de

lipid charge[12,19]. Advanced microscopy techniques like hyperspectral imaging and fluorescence lifetime imaging microscopy (FLIM) combined with the phasor analysis[20,21], have allowed to obtain quantitative information on the condensate-membrane interaction at the nanoscale. These approaches revealed a general mechanism by which condensates can locally increase the lipid packing and dehydration depending on their affinity for the membrane[22]. Altogether, these observations constituted systematic studies addressing the mechanisms of interaction between condensates and membranes. However, due to the vast diversity of condensates and the different conditions under which they interact, many questions remain unanswered.

We distinguish two classes of membrane-condensate systems extensively explored in the literature. In the first one, the protein molecules are tethered to the membrane through specific lipid binding such as with NTA (Ni-nitrilotriacetic-acid) lipids, and liquid-liquid phase separation (LLPS) takes place at the membrane surface, producing two-dimensional (2D) condensates[13,23]. This association allows the condensate to colocalize with a specific lipid phase (domain) in phase separated membranes[6]. In the second class, in which bio(macro)molecules form three-dimensional (3D) condensates on their own, the association with a specific lipid phase can also be driven via a lipid anchor[24]. For example, model polymer-based condensates formed through LLPS of solutions of polyethylene glycol (PEG) and dextran (also known as aqueous two-phase system, or ATPS) have been shown to induce phase separation in membranes containing PEGylated lipids as tethers[24–26]; the wetting by the droplets was found to lead to vesicle budding and lateral redistribution of the lipids that matched the droplet-induced budding pattern[26]. In addition, it has been suggested that the lipid phase state drives the phase specific binding of non-tethered 3D condensates[27]. Some studies show that organization in the membrane is altered by interactions with crowded solutions of proteins and polymers due to changes in the activity of the interfacial water[28,29]. However, a systematic evaluation of the effect of membrane packing on condensate-membrane interactions and how membrane organization influences wetting by biomolecular condensates is missing.

To address this, here we evaluated the effect of lipid chain length and cholesterol content on the interactions with non-tethered 3D condensates. Our aim is to determine whether lipid packing impacts condensate-membrane affinity in the absence of specific anchors. To assess the membrane fluidity, we utilized LAURDAN, a lipid-like fluorescent dye sensitive to polarity changes and water dipolar relaxation in the membrane, designed over forty years ago by Weber and Farris[30,31]. To date, LAURDAN remains one of the most sensitive fluorescent probes for detecting changes in membrane packing and hydration, and it is extensively used in both in vitro and in vivo studies[32]. Traditionally, LAURDAN spectral changes in membranes have been quantified using a ratiometric analysis of the two main emission bands, known as generalized polarization (GP), providing a measure for the physical state of the membrane[31–33]. In the last decade, the spectral phasor approach, which involves taking the Fourier transform of the whole spectrum[34], has further exploited the properties of LAURDAN, broadening its applications for microscopy and cuvette experiments[20–22,32,35–37]. We combined hyperspectral imaging with phasor analysis to quantify changes in membrane fluidity. Using microscopy images and theoretical analysis, we determined the condensate affinity for the membrane and the membrane interfacial tensions. The combination of these two approaches enabled us to establish a fluidity scale that correlates changes in packing with corresponding variations in condensate affinity.

Our findings demonstrate that, in the absence of specific interactions or lipid anchors, lipid packing regulates the wetting affinity of condensates. Increasing lipid chain length or cholesterol content decreases the condensate-membrane interaction. In contact with phase-separated membranes, this mechanism drives the condensate specificity for a given domain. Moreover, the protein affinity for the membrane can induce the formation of tubes and double membrane-sheets by altering the membrane spontaneous curvature. Extending our results to condensate systems with diverse material and electrical properties, suggests that membrane order can generally regulate wetting by biomolecular condensates.

## Results

### Utilizing LAURDAN spectral phasors to finely measure membrane packing changes

Hyperspectral Imaging (HSI) is a microscopy technique that captures a stack of images where each pixel contains spectral information. These data can be analyzed using the spectral phasor approach, which applies a Fourier transform to produce a vector (phasor) for each pixel in a polar plot called the spectral phasor plot[34]. The angular position of the phasors corresponds to the center of mass of the emission spectra, while the radial position relates to the spectral width[38].

Figure 1a shows spectra for LAURDAN in DOPC, DLPC and DPPC giant unilamellar vesicles (GUVs), acquired with HSI at room temperature (23 °C). These phospholipids share the same polar headgroup (phosphocholine) but differ in hydrocarbon chain length and saturation: DOPC has two 18-carbon chains with one double bond each, while DLPC and DPPC have saturated chains with 12 and 16 carbons, respectively. This results in membranes with varying degrees of lipid packing and hydration: for the same headgroup, increasing the chain length enhances the van der Waal interactions, reducing the area occupied by each lipid, and enhancing the shielding of the hydrophobic bilayer core. In Fig. 1a it is evident that both the position and shape of the LAURDAN emission spectrum vary with different membrane compositions, as previously reported[28]. The spectrum for DOPC is shifted to longer wavelengths due to the lower degree of lipid packing and the higher water dipolar relaxation. In contrast, the DPPC membrane is highly packed and exhibits reduced dipolar relaxation as water dynamics around the LAURDAN moiety is limited[31]. The DLPC spectrum falls between these extremes, with an intermediate degree of lipid packing and hydration, as expected[33,39]. While DOPC and DLPC are in the fluid phase (liquid disordered, $L_d$) at 23 °C, below their melting temperatures ($T_m$), DPPC ($T_m = 41$ °C) is in the gel phase. The spectral shift between DPPC and DOPC membranes is ~50 nm, one of the highest reported for membrane solvatochromic dyes[40,41], highlighting LAURDANs sensitivity to subtle changes in membrane packing and hydration[32,33,39].

Figure 1b shows an example of a phasor plot for GUVs made of these different lipids. In this plot, the increased lipid packing is seen as a clockwise displacement of the phasor clouds[21,35,36,42]. Since each pixel in HSI contains spectral information, the phasor transformation results in a pixel cloud with coordinates corresponding to the spectrum shape and position. One advantage of the phasor approach is the ability to exploit the linear algebra of the Fourier space[43]. Due to the linear combination properties of the Fourier space, membranes with varying degrees of packing and hydration form a linear trajectory in the spectral phasor plot, as observed in Fig. 1b. The extremes of this trajectory correspond to distinct surrounding environments for LAURDAN, reflecting various degree of water penetration in the lipid bilayer[32,36,42].

It is important to highlight that the changes in polarity due to packing differences and water dipolar relaxation cannot be separated using the LAURDAN spectrum. Therefore, the term fluidity is used here to describe changes in both parameters[21,32]. The extremes of the linear trajectory can be defined as the phasor positions for gel and liquid phases, or can be arbitrarily defined, as done here. Using the two-component analysis (see Methods), the pixel distribution along the defined trajectory can be obtained, allowing quantification of differences in the lipid packing and hydration (fluidity), as shown in Fig. 1c.

One of the most intriguing features of the phasor approach is the reciprocity principle[21,44]. This principle allows for selecting pixels in the

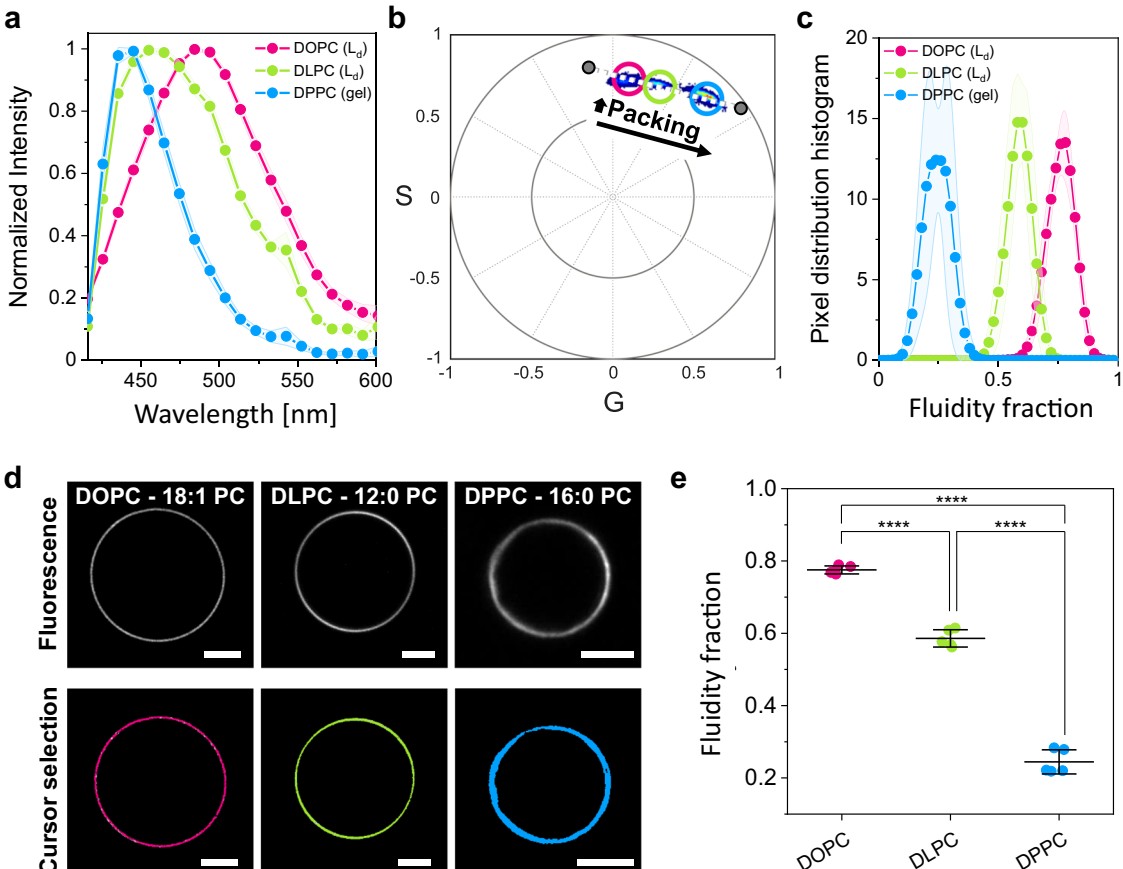

**Fig. 1 | LAURDAN spectral phasors allow measurement of changes in membrane packing and hydration. a** LAURDAN spectra reconstituted from hyperspectral imaging of GUVs made of three different lipids. Data are represented as the mean (dots and lines) ± SD (shadowed contour), $n = 5$ vesicles per condition. **b** Spectral phasor plot for hyperspectral images of DOPC, DLPC, and DPPC GUVs containing 0.5 mol% LAURDAN at $(23 \pm 1)°$C. The plot corresponds to at least five images per condition. Increasing the chain length results in an increase in packing evidenced by the shift of the pixel clouds in a clockwise manner. The pixel clouds are colored according to the pixel density, increasing from blue to red. **c** Pixel distribution histograms along the linear trajectory (white dotted line in **b**), showing the fluidity

fraction for the different lipid membranes. Data are represented as the mean (dots and lines) ± SD (shaded area), $n = 5$ independent experiments per condition. **d** Representative confocal microscopy images of GUVs of the indicated lipids (upper panel). Using circular cursors to select the pixel clouds in (**b**), the corresponding pixels are colored in the images, as shown in the lower panel. Scale bars: 5 μm. **e** Center of mass of the histograms shown in (**c**). Individual data points are shown for each membrane composition. The lines indicate the mean value ± SD ($n = 5$). The statistical analysis was performed with One-way ANOVA and Tukey post test analysis ($p < 0.0001$, **** | $p < 0.001$, *** | $p < 0.01$, ** | $p < 0.05$, * | ns non-significant). Source data are provided as a Source Data file.

phasor plot with cursors (as those shown in Fig. 1b), which in turn colors the corresponding pixels in the image, as exemplified in Fig. 1d. This creates a visual connection between the spatial information and the spectral changes.

To perform statistical analysis on these changes, we calculate the center of mass of the histograms shown in Fig. 1c, as shown in Fig. 1e. This provides a precise and sensitive measurement of the physical state of the membrane in terms of lipid packing and hydration.

In the following, we utilize LAURDAN spectral phasors to correlate changes in membrane fluidity with the wetting behavior of biomolecular condensates.

## Membrane lipid packing determines the wetting affinity of biomolecular condensates

Biomolecular condensates have been shown to interact and remodel membranes depending on the salinity and the membrane composition[12,45,46]. Wetting by biomolecular condensates can locally influence membrane packing and hydration, offering a mechanism to modulate membrane properties through regulating the degree of wetting[22]. Here, we investigated whether variations in membrane packing could reciprocally affect condensate wetting under the same buffer conditions. Using glycinin, a soybean protein known to phase separate in response to salinity[47], we examined its interaction with

three different membrane compositions: DOPC, DLPC, and DPPC. All experiments were conducted under identical working conditions: 150 mM NaCl at 23 °C, with a protein concentration of 10 mg/mL. Figure 2a illustrates that DOPC membranes exhibit nearly complete condensate spreading, consistent with previous findings[12]. In contrast, DLPC and DPPC membranes display distinct wetting morphologies, underscoring the influence of lipid packing on condensate wetting.

Membrane wetting by biomolecular condensates is quantified by the contact angles formed at the intersection of the two membrane segments and the condensate surface[45,48], as illustrated in Fig. 2b. Note that these contact angles are apparent, can vary between vesicle-condensate couples, and do not reflect the geometry at the nanometer scale, where the membrane is smoothly curved rather than exhibiting a sharp kink[49]; at this scale, wetting is characterized by an intrinsic contact angle $\theta_e^{in}$[50], as shown in Fig. 2b. The three apparent microscopic angles, $\theta_i$, $\theta_e$, and $\theta_c$, are related to the three interfacial tensions $\Sigma_{ie}^m$, $\Sigma_{ic}^m$, and $\Sigma_{ce}$, which are balanced at the vesicle-condensate contact line, forming the sides of a triangle (Fig. 2b)[48]. This allows us to introduce the geometric factor $\Phi = (\sin\theta_e - \sin\theta_c)/\sin\theta_i$[12,45,48], a dimensionless quantity that depends only on the material properties of the membrane and the condensate; indeed $\Phi = \cos\theta_{in}$[46]. $\Phi$ takes extreme values of $\Phi = 1$ (or $\theta_e^{in} = 0°$) for dewetting, and $\Phi = -1$ (or $\theta_e^{in} = 180°$) for complete wetting (see Methods for details)[12,45]. In order to obtain the

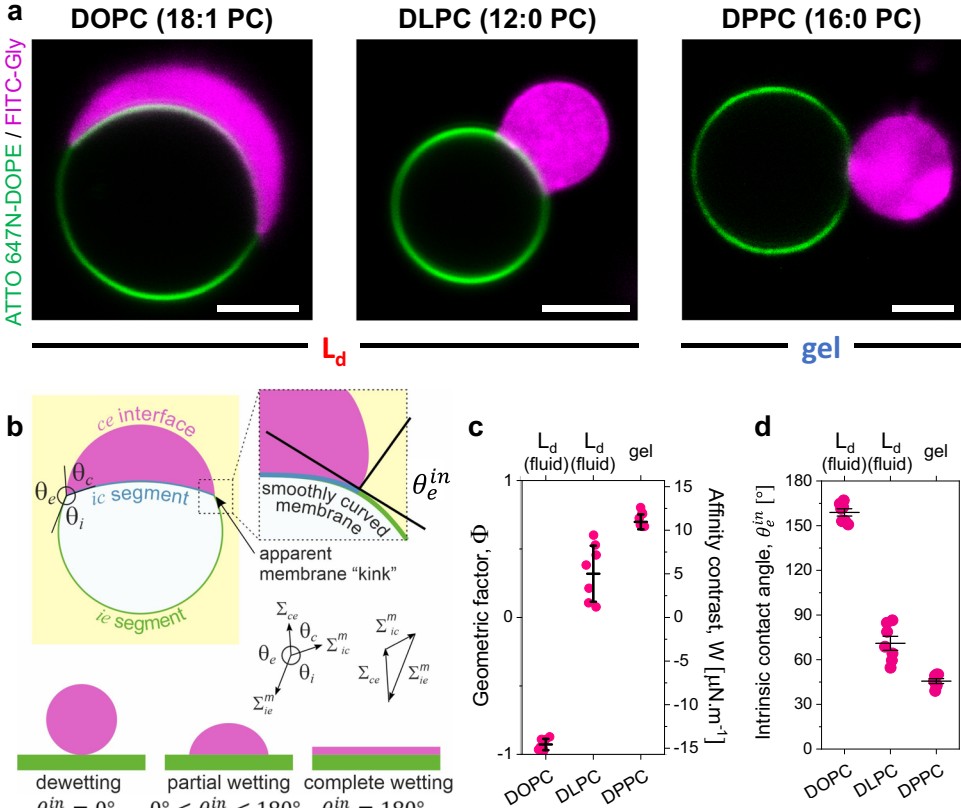

**Fig. 2 | Membrane packing determines the wetting affinity of biomolecular condensates. a** Representative confocal microscopy images of GUVs of the indicated lipids labeled with 0.1 mol% ATTO 647N-DOPE (green) in contact with glycinin condensates labeled with FITC-glycinin (magenta) at working conditions (150 mM NaCl, protein concentration 10 mg/mL, 23 ± 1 °C). Scale bars: 5 μm. **b** Sketch showing the parameters that define the geometric factor (Φ). The three apparent contact angles $\theta_i$, $\theta_e$, and $\theta_c$ (observed microscopically), facing the vesicle interior (*i*), the external solution (*e*) and the condensate (*c*), are related to the three interfacial tensions occurring in the system, $\Sigma_{ce}$, $\Sigma_{ie}^m$, and $\Sigma_{ic}^m$. The three tensions are balanced at the three-phase contact line (black circle) forming the sides of a triangle[45,48], as shown on the right. At the nanoscale, the membrane is smoothly curved and wetting is characterized by the intrinsic contact angle $\theta_e^{in}$; which is

defined as $\cos\theta_e^{in} = (\sin\theta_e - \sin\theta_c)/\sin\theta_i$. The value of $\theta_e^{in}$ varies between 0° and 180° depending on the affinity of the condensate droplet for the membrane, as shown in the bottom in analogy to the behavior of liquid droplets at solid substrates. **c** Geometric factor ($\Phi = (\sin\theta_e - \sin\theta_c)/\sin\theta_i$, left axis) and affinity contrast ($W = \Sigma_{ce}/\Phi$, right axis) for the different membrane compositions characterizing the affinity of the condensate to the membrane. Individual data points are shown for each membrane composition. The lines indicate the mean value ± SD (*n* = 7). **d** Intrinsic contact angle $\theta_e^{in} = \arccos\Phi$[45,48] for the different membrane compositions. Individual data points are shown for each membrane composition. The lines indicate the mean value ± SD (*n* = 7). Source data are provided as a Source Data file.

correct values of the apparent contact angles, one must use the equatorial cross-sections for both the vesicle and the droplet, as explained in detail previously[12]. It has been demonstrated that there are no significant differences between deriving the intrinsic contact angle from the apparent ones using optical microscopy or directly measuring it using super-resolution microscopy[49]. Figure 2c shows that increasing the degree of lipid packing drives dewetting.

It can be shown that the geometric factor, Φ, is equal to the rescaled affinity contrast, $W/\Sigma_{ce}$, which is a mechanical quantity that describes the different adhesion free energies per unit area of the two membrane segments[12,45], see Methods for details. The affinity contrast, W, compares the membrane affinity for the condensate versus the affinity for the protein-poor phase, taking negative values when the membrane prefers the condensate and positive values when it prefers the external buffer. By measuring the condensate interfacial tension $\Sigma_{ce}$, it is possible to deduce the affinity contrast, W, which can range from μN/m to mN/m. For glycinin condensates under the conditions used here, the interfacial tension $\Sigma_{ce}$ = 15.7 μN/m, as determined from rheology and condensate coalescence measurements[22]. Figure 2c shows that the affinity contrast, W, increases when increasing lipid packing, indicating that the membrane prefers the external buffer over the condensate. In Fig. 2d, the intrinsic contact angle obtained from the apparent microscopic contact angles is plotted for the different

membrane compositions, showing that $\theta_e^{in}$ decreases for increased packing, i.e. dewetting becomes more favorable. Note that, contrary to the observed contact angles, the affinity contrast W, the interfacial tension is $\Sigma_{ce}$, the geometric factor Φ, and the intrinsic contact angle $\theta_e^{in}$, are material parameters, which are independent of the size and shape of the chosen condensate-vesicle couple[11,12,45].

## Cholesterol content modulates condensate wetting

To further demonstrate the influence of lipid packing on interactions with biomolecular condensates, we prepared GUVs with varying cholesterol fractions. Cholesterol impacts several membrane properties, such as lipid packing and hydration[51], permeability and compressibility[52], and bending rigidity[53]. It is also crucial for forming domains or rafts that promote receptor clustering in cell signaling[6,54].

Figure 3a shows the phasor plot for LAURDAN in DOPC GUVs with varying cholesterol (Chol) levels. Increasing cholesterol enhances lipid packing, again placing the data on a linear trajectory, as expected. Figure 3b, c shows the quantification of the fluidity fraction for the DOPC:Chol mixtures. Note that the linear trajectory in Fig. 3a aligns with that in Fig. 1b, as we fixed the extreme points to allow comparison across different data. The upper panel in Fig. 3d displays vesicles colored according to the pixels selected in Fig. 3a. Overlapping pixel clouds and circular cursors can cause vesicles to be painted with

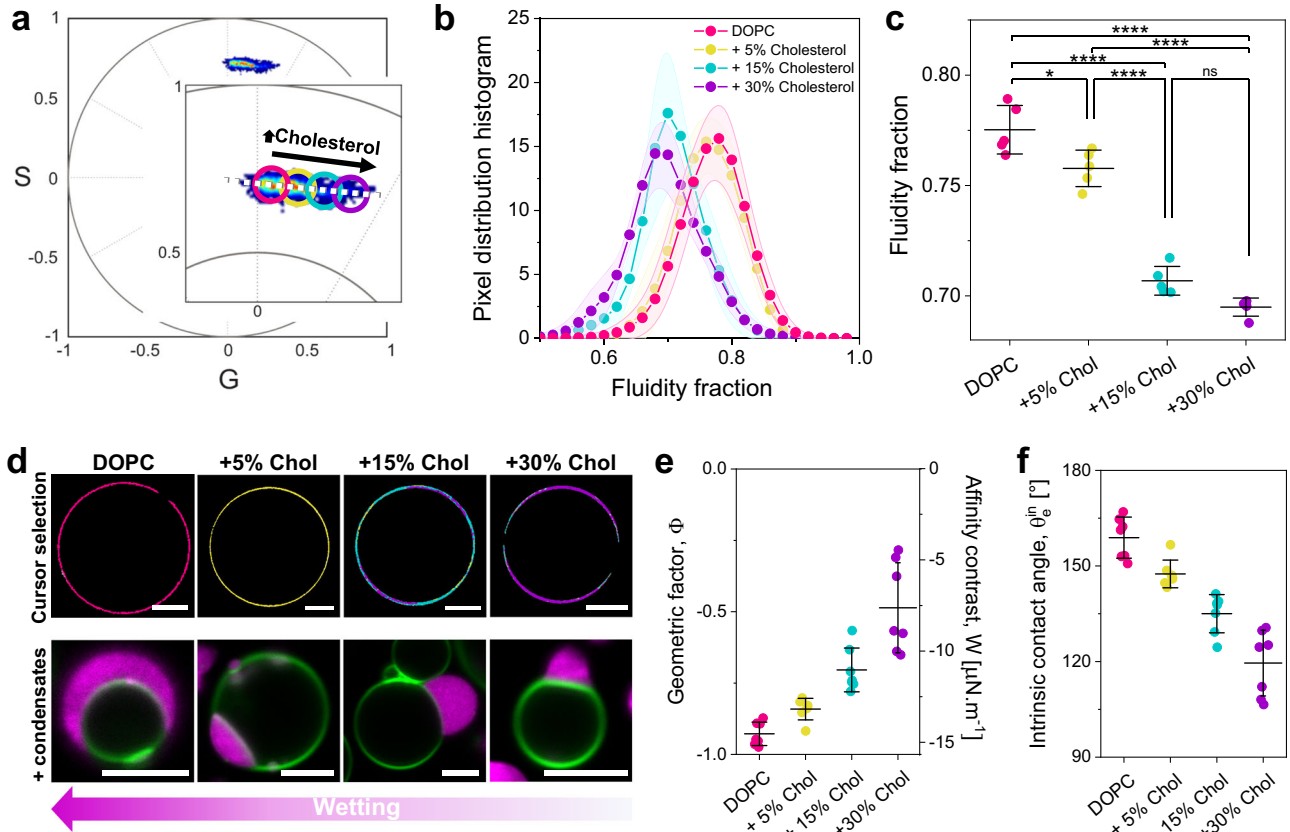

**Fig. 3 | Cholesterol-induced lipid packing modulates condensate wetting.**
**a** Spectral phasor plot for GUVs made of DOPC and 0%, 5%, 15%, and 30% mol of cholesterol and containing 0.5 mol% LAURDAN at (23 ± 1)°C. **b** Pixel distribution histogram along the linear trajectory drawn as a white dotted line in (**a**), showing the fluidity fraction for the different membrane compositions. Data are represented as the mean (dots and lines) ± SD (shadowed contour), $n = 5$ independent experiments per condition. **c** Center of mass of the histograms shown in (**b**). Individual data points are shown for each membrane composition. The lines indicate the mean value ± SD ($n = 5$). The statistical analysis was performed with One-way ANOVA and Tukey post-test analysis ($p < 0.0001$, **** | $p < 0.001$, *** | $p < 0.01$, ** | $p < 0.05$, * | ns = non-significant). **d** Upper panel: cursor colored images of GUVs

of the indicated compositions, corresponding to the cursors shown in (**a**). Lower panel: representative confocal microscopy images of GUVs labeled with 0.1 mol% ATTO 647N-DOPE (green) of the indicated compositions in contact with condensates labeled with FITC-glycinin (magenta) at the working conditions. Scale bars: 5 μm. **e** Geometric factor and affinity contrast for condensate/membrane systems at the different membrane compositions. Individual data points are shown for each membrane composition. The lines indicate the mean value ± SD ($n = 5$). **f** Intrinsic contact angle for condensate/membrane systems at the different membrane compositions. Individual data points are shown for each membrane composition. The lines indicate the mean value ± SD ($n = 5$). Source data are provided as a Source Data file.

multiple colors; this should not be confused with domain formation, as all of these mixtures are homogeneous and in the liquid-disordered phase. The lower panel in Fig. 3d shows vesicles with different cholesterol content in contact with glycinin condensates. As cholesterol content increases, the condensate affinity for the membrane (wetting) decreases, quantified by the geometric factor and affinity contrast in Fig. 3e, and the intrinsic contact angle in Fig. 3f.

While DOPC:Chol membranes are in the liquid-disordered phase ($L_d$), adding cholesterol to DPPC results in the liquid-ordered ($L_o$) phase. Cholesterol increases membrane packing when mixed with unsaturated lipids like DOPC, as shown in Fig. 3, but fluidizes membranes made of saturated lipids like DPPC, as shown in Supplementary Fig. 1, comparing DPPC and DPPC:Chol 70:30. For these compositions, there are no significant differences in condensate wetting, since both the $L_o$ and gel phases are highly packed, and the geometric factor is near the limit for complete dewetting ($\Phi = 1$, $\theta_e^{in} = 0°$).

### Lipid packing governs phase-specific interaction in phase-separated membranes

The affinity of a condensate for a lipid phase can be modulated by specific tethers for both 2D[6,55] and 3D[24] condensates. However, for non-tethered 3D condensates (studied here), preferential lipid phase binding has been attributed to the phase state[27], and has been

observed as droplet-mediated budding for phase separation of polymer mixtures inside GUVs[26]. Above, we demonstrated that condensate affinity can be regulated solely by the membrane packing rather than the phase state, without the need for specific tethers or charges. Liquid-disordered ($L_d$) phases showed high and intermediate affinity for the glycinin condensates (Figs. 2, 3), while gel and liquid-ordered ($L_o$) phases showed much lower affinity, near dewetting (Fig. 2 and Supplementary Fig. 1). To test whether these affinity differences could drive condensate specificity for a given phase in phase-separated membranes, we prepared GUVs of DOPC:DPPC 1:1 displaying fluid/gel phase coexistence (see Supplementary Fig. 2), and exposed them to condensates. The fluorescent membrane label (ATTO647N-DOPE) partitions to the fluid phase, making the gel phase appear black in fluorescence microscopy images. Figure 4a, b shows that condensates only interact with the fluid phase, avoiding the gel phase. Similarly, in ternary mixtures of DOPC:DPPC:Chol (1:1:1) displaying liquid-disordered/liquid-ordered ($L_d/L_o$) phase separation, condensates only interact with the liquid-disordered phase (Fig. 4c). To isolate the effect of lipid packing and minimize chemical changes at the interface, we have kept the phospholipid headgroup constant (choline). However, when substituting DPPC with sphingomyelin (SM) to form phase-separated GUVs of the canonical ternary mixture DOPC:SM:Chol 1:1:1[56], we observe the same behavior: condensates only interact with the

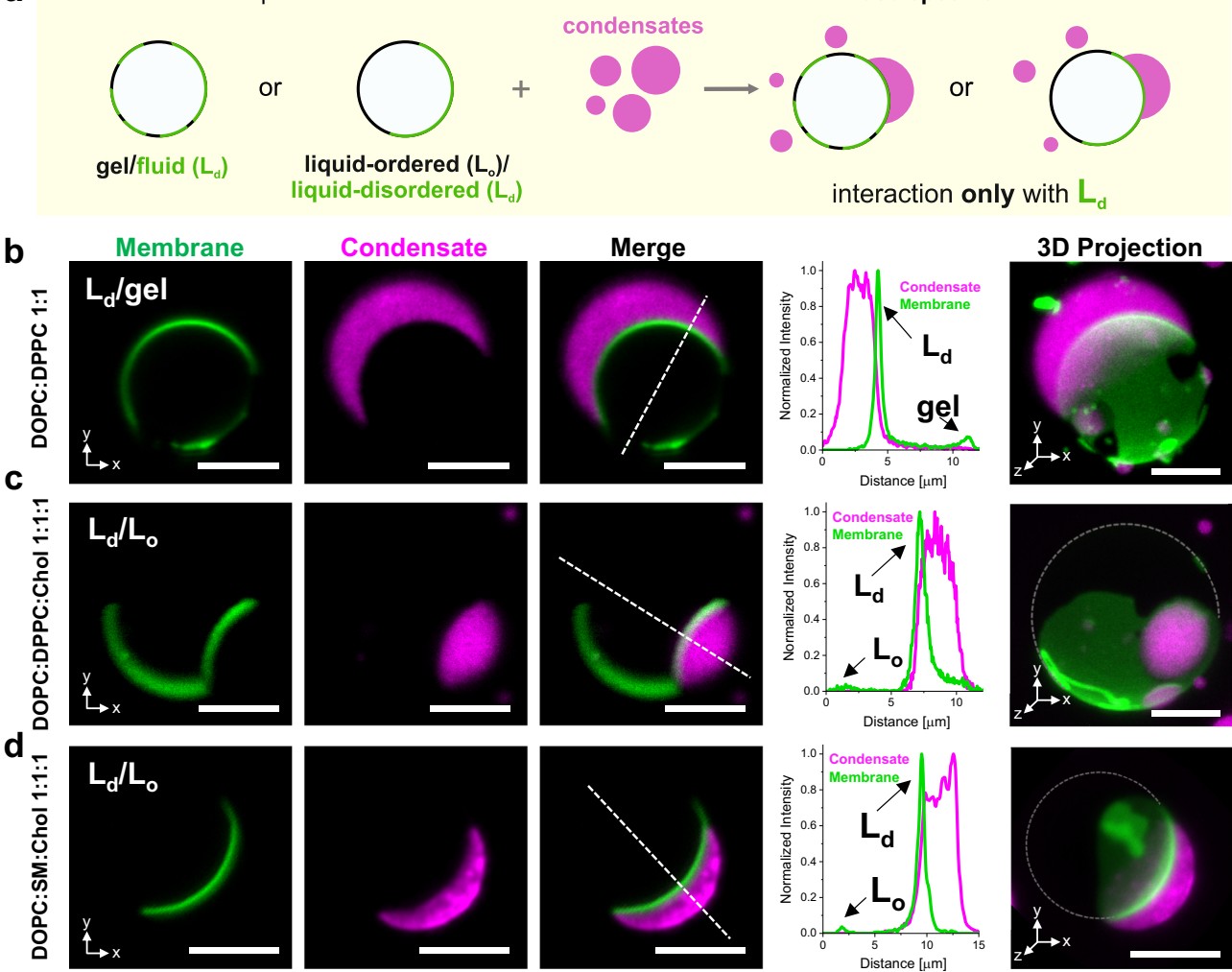

**Fig. 4 | Lipid packing determines wetting phase specificity in phase-separated membranes. a** Sketch illustrating that in the presence of gel/fluid ($L_d$) or liquid-ordered ($L_o$)/liquid-disordered ($L_d$) phase-separated GUVs, the condensates bind exclusively to the liquid-disordered ($L_d$) phase. **b–d** Confocal sections (x,y), line profiles, and 3D projections (x,y,z), of phase-separated GUVs showing that glycinin condensates (magenta) only interact with the liquid-disordered phase (green), excluding the gel or the liquid-ordered phase. Scale bars: 5 μm. GUVs were labeled with 0.1 mol% ATTO647 N-DOPE and the lipid compositions are: (**b**) DOPC:DPPC 1:1;

(**c**) DPPC:DOPC:Chol 1:1:1 and (**d**) DPPC:SM:Chol 1:1:1; cross-sections and 3D projections correspond to the same vesicle-condensate couple with the specific membrane composition. The line profiles show that condensates are always interacting with the membrane segments of highest intensity which corresponds to the $L_d$ phase. Dashed lines in the 3D projections are a guide to the eye indicating the vesicle contour. See also Supplementary Movies 1–3. All images were taken under the working conditions defined above. Source data are provided as a Source Data file.

liquid-disordered phase, as shown in Fig. 4d and Supplementary Fig. 2. This result suggests that the effect of lipid packing on the condensate affinity is independent of the lipid type. Larger field-of-view images (Supplementary Fig. 2) showing several vesicle-condensates pairs confirm that condensates only wet the liquid-disordered phase, excluding the gel or liquid-ordered phase, respectively, for both binary and ternary lipid mixtures.

The degree of lipid packing (fluidity fraction) of the phases in coexistence explains this behavior, as shown in Supplementary Fig. 3: in the binary mixture DOPC:DPPC 1:1 the fluid phase is close to that of pure DOPC, while the gel phase is close to pure DPPC. In the ternary mixture containing DPPC, the liquid-disordered phase has a fluidity fraction between that of DOPC:Chol 7:3 and DLPC, and the liquid-ordered phase lies close to DPPC:Chol 7:3 (Supplementary Fig. 3). Upon replacing DPPC with SM, both the liquid-disordered and liquid-ordered phases exhibit higher fluidity compared to the ternary mixture containing DPPC, as shown in Supplementary Fig. 3. These results demonstrate that in the absence of specific tethers or electrostatic interactions, condensate specificity for a

given lipid phase is primarily determined by the degree of lipid packing.

## Effect of membrane composition and bending rigidity on membrane remodeling by biomolecular condensates

Biomolecular condensates can remodel membranes[12,46], which is crucial in many cellular processes[7,9,14]. Glycinin condensates induce interfacial ruffling, forming undulations and finger-like protrusions[12], similar to the protein pockets observed in plant tonoplasts[57]. This ruffling depends on the available excess area and can be modulated by tension[12]. In general, GUV suspensions are heterogeneous in terms of initial tension and membrane excess area. Under the working conditions used here, we also observed tubulation in approximately one third of the vesicles within a sample. Figure 5a, b shows nanotubes forming at the membrane-condensate interface and protruding into the condensate phase (see 3D projections in Supplementary Fig. 4 and Supplementary Movies 4 and 5). For DOPC vesicles, the tube diameters are below the optical resolution (Fig. 5a), while the phase separated DOPC:DPPC 1:1 membrane shows pearled-like tubes with dimensions

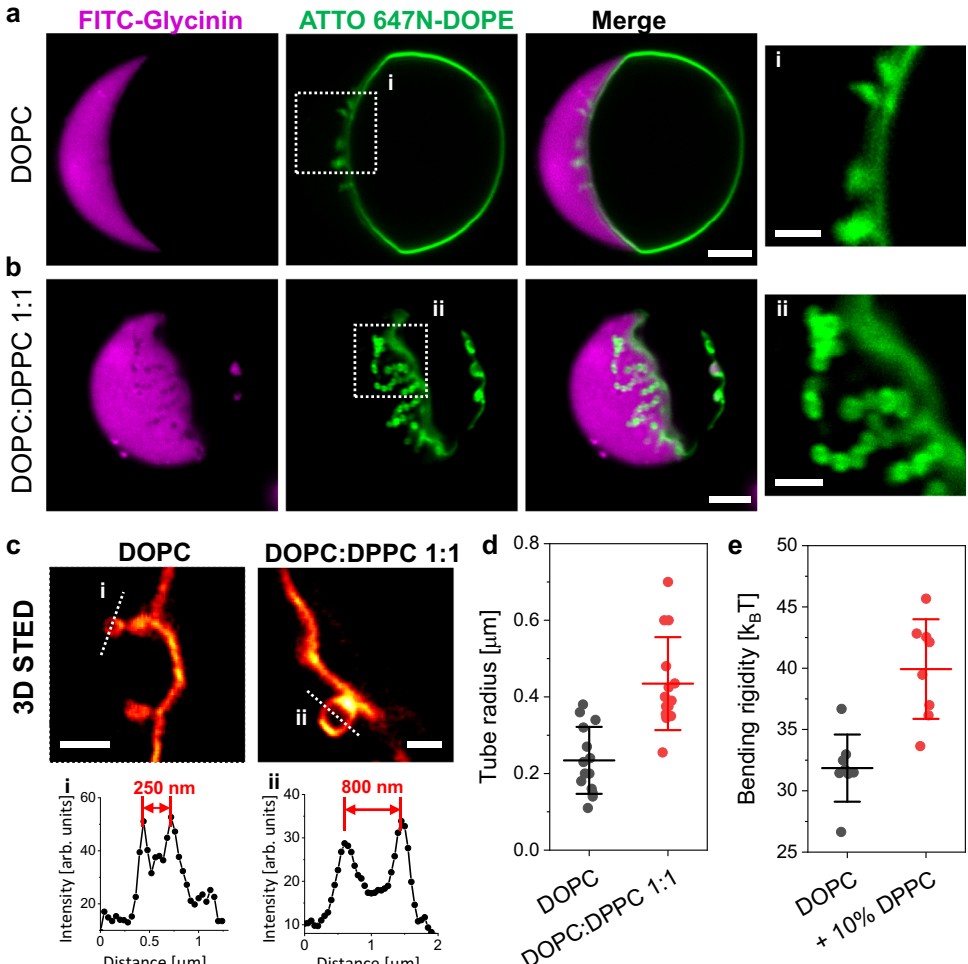

**Fig. 5 | Nanotubes form at the membrane-condensate interface with diameters that depend on bending rigidity.** Confocal microscopy images of DOPC (**a**) and DOPC:DPPC 1:1 (**b**) giant vesicles in contact glycinin condensates displaying nanotube formation at the membrane-condensate interface at working conditions. The last panels (i, ii) show the zoomed regions indicated in the membrane channel. Scale bars: 5 μm, zoomed images: 2 μm. **c** 3D STED imaging allows resolving the tube morphology and dimensions. Scale bars: 1 μm. **d** Tube radius measured from STED images of DOPC and DOPC:DPPC 1:1. Individual measurements are shown as dots and the lines indicate mean ± SD (*n* = 14). **e** Membrane bending rigidity measured by fluctuation spectroscopy for DOPC and DOPC:DPPC 9:1. Individual measurements are shown as dots and the lines indicate mean ± SD (*n* = 8). Source data are provided as a Source Data file.

within the optical resolution (Fig. 5b). Stimulated emission depletion (STED) super-resolution microscopy reveals average tubes radii of $(0.23 \pm 0.09)$ μm for DOPC and $(0.43 \pm 0.12)$ μm for DOPC:DPPC 1:1 membranes (Fig. 5c, d). Note that due to the curved interface, STED images only show tubes in the focal plane; see Supplementary Movies 6, 7 for STED microscopy z-stacks showing tubes at different planes.

The spontaneous formation of inward tubes has been previously observed in the PEG-rich phase of GUVs encapsulating PEG/dextran ATPS[58,59]. Tubes nucleate from small buds, grow into necklace-like structures, and can become cylindrical above a critical length[59]. When comparing $L_d$ and $L_o$ membranes, tube diameter depends on the bending rigidity; higher bending rigidity results in higher tube diameter[13,59].

Here, we observed outward-protruding tubes from the fluid phase of phase-separated DOPC:DPPC 1:1 membranes into the condensates, showing larger diameter than those in pure DOPC. Considering that the fluid phase in DOPC:DPPC 1:1 membranes contains about 10% of DPPC[60], we measured and compared the bending rigidity of DOPC and DOPC:DPPC 9:1 by fluctuation spectroscopy[61]. We chose this binary mixture because it is homogeneous and similar in composition to the phase from which the tubes protrude. Figure 5e shows that the bending rigidity of the binary mixture is at least 1.2 times higher than

for pure DOPC, which could explain the observed diameter difference. These results show that by tuning the membrane composition and therefore the bending rigidity, non-tethered condensates can induce nanotube protrusions of different thickness. The process is similar to that observed in vesicles encapsulating PEG/dextran ATPS[59], or exhibiting two-dimensional tethered condensates[13].

## Nanotube and double-membrane sheet formation driven by protein adhesion and spontaneous curvature

Nanotube formation is generally stabilized by spontaneous curvature[48,58] that can be generated by various factors producing an asymmetry across the bilayer[62]. For example, the presence of different ions, soluble molecules or pH across the bilayer[63–65], lipid asymmetry[66], or the adsorption of polymers or proteins to only one leaflet can cause tubulation[62]. Since the tubes formed in the presence of condensates always protrude into the condensate phase (Fig. 5), we tested whether this tubulation was due to spontaneous curvature generated from protein adsorption. To probe the effect of protein adsorption to the membrane excluding the effect of the bulk condensate phase, it is necessary to work under conditions in which the protein solution is homogeneous and condensates are not formed (i.e. away from the two-phase coexistence region outlined by the binodals). Figure 6a shows the phase diagram for glycinin[47], indicating the conditions for

condensate formation (here, 150 mM NaCl) and two conditions in which glycinin presents a homogeneous solution—at low (20 mM) and high (365 mM) NaCl concentrations.

In most of our experiments, vesicles were grown in sucrose and then diluted in an isotonic NaCl solution (see Methods). To determine whether the observed tubulation could also arise from this solution asymmetry, we analyzed vesicles in the absence of the protein. As shown in Fig. 6b, sucrose/NaCl asymmetry alone induces inward tubulation, consistent with previous findings[63]. However, because the tubes point inwards, reflecting negative spontaneous curvature, this result rules out solution asymmetry as the cause of outward tubulation (positive spontaneous curvature) observed in Fig. 5. Instead, the outward tubulation must arise from other factors such as protein adsorption.

To assess the protein effect on the membrane, we prepared vesicles in the presence of homogeneous glycinin solutions at low and high NaCl concentrations (as indicated in Fig. 6a). These vesicles were directly grown in the NaCl solutions of desired concentration, to avoid solution asymmetry across the membrane. Figure 6c, d shows that glycinin in homogeneous solution adsorbs on the membrane, forming outward buds and tubes. The protein signal at the membrane increases with salinity (Fig. 6e) and is associated with more extensive tubulation at 365 mM compared to 20 mM NaCl. The increased adsorption at higher salinity aligns with previous observations on affinity of glycinin condensates to membranes[12] and is corroborated by mass photometry data on supported lipid bilayers, showing a two-fold increased adsorption as shown in Fig. 6f. The latter data, obtained with label-free protein, eliminate potential artifacts related to quantum yield variations in fluorescence intensity measurements and indicate enhanced adsorption at higher salinity. At 20 mM NaCl, glycinin adsorbs predominantly as trimer (160 kDa) and additionally as hexamer (320 kDa), while at 365 mM NaCl it also adsorbs as nonamer (480 kDa) complexes (Supplementary Fig. 5).

Interestingly, we observed that upon extensive tubulation (occurring when higher excess area is available), outward tubes can adhere to the GUVs, form branches over time, and transform into double-membrane sheets (Fig. 6g, h). In GUVs encapsulating PEG/dextran ATPS, nanotubes adsorbed at condensate interfaces have been shown to transform into cisterna-like double-membrane sheets, a wetting driven process, dependent on the interfacial tension and spontaneous curvature[67]. Here, we observe double-membrane sheets adsorbing onto the GUV covered by protein rather than onto a condensate surface, suggesting that the structures are stabilized by protein-mediated membrane-membrane adhesion. Note that both nanotubes and double-membrane sheets adhere to the GUV surface, making them difficult to clearly distinguish from confocal microscopy cross-sections. Visualization of double-membrane sheets requires z-stacks for 3D projections or STED imaging (as shown in Fig. 6g, Supplementary Fig. 6, and Supplementary Movie 8). Overall, these results indicate that protein adsorption to the bilayer can generate spontaneous curvature stabilizing nanotubes and double-membrane sheets.

## Correlation between condensates wetting affinity and membrane lipid packing extends to condensate systems with different properties

Glycinin is a hexamer of high molecular weight (360 kDa)[68], and its phase diagrams have been determined for different conditions by varying protein concentration, pH, salinity, and temperature[47]. This makes glycinin a very convenient model protein for studying membrane-condensates interactions under different conditions[12,22]. Glycinin contains a hypervariable, intrinsically disordered region (IDR) of low complexity, rich in aspartate and glutamate residues which is believed to promote phase separation[47]. Moreover, salt-triggered glycinin phase separation proceeds with an increase in random-coil

motifs[22]. While the mechanism of interaction between non-anchored condensates and membranes is still poorly understood[46], the membrane wetting by glycinin condensates is likely to be mediated by hydrophobic interactions, because charged membranes promote dewetting[12]. To determine whether the dependence of wetting affinity on membrane lipid packing applies broadly rather than being specific to glycinin, we extended our study to other condensate systems with different chemical and material properties. These include: (i) condensates formed by the neutral polymers PEG and dextran, (ii) condensates formed by the full-length intrinsically disordered protein Synapsin 1 (Syn1), and (iii) condensates formed by two oppositely charged oligopeptides.

Condensates formed by mixtures of PEG and dextran have been extensively studied and are a hallmark of segregative phase-separation[46,69]. These condensates exhibit ultralow interfacial tension and low viscosity compared to most protein- or peptide-based condensates[70] (see summary of material properties in Supplementary Table 1). Despite the neutral nature of PEG and dextran and their minimal interaction with membranes[69], PEG/dextran condensates can induce extensive membrane remodeling[46]. Pioneer experiments done in ATPS demonstrated that condensates can wet and remodel membranes[18,71]. As shown in Fig. 7a, when PEG/dextran condensates are brought into contact with vesicles of increasing lipid packing, the condensate/membrane affinity decreases, following a similar trend to that observed for glycinin (Fig. 2).

Next, we tested the interaction of membranes with the full-length protein Syn1. Syn1 is the most abundant synaptic phosphoprotein and it contains a large IDR (a.a. 416-705) that has been shown to be necessary and sufficient for triggering phase separation in vitro[72,73]. Syn1 condensates have low affinity for neutral membranes, but their interaction can be significantly enhanced by incorporating negatively charged lipids[74]. Thus, to test how lipid packing affects Syn1 condensate-membrane affinity, it was essential to begin with conditions where the condensate-membrane interaction is robust for membranes with low lipid packing. For this reason, to enhance condensate-membrane interaction, we prepared GUVs made of DOPC, DLPC, and DPPC with 10 mol% DOPS (all forming homogeneous membranes). The phasor plot and fluidity fraction histograms are shown in Supplementary Fig. 7. Inclusion of the charged DOPS increased membrane fluidity (Supplementary Fig. 8) and reduced the fluidity difference between DOPC and DLPC (compare Fig. 1c and Supplementary Fig. 7c). Figure 7b shows that when Syn1 condensates are in contact with charged GUVs of increasing lipid packing, the condensate-membrane affinity decreases, further corroborating our findings.

Finally, we tested a system presenting heterotypic and associative phase separation. The oligopeptides poly-L-lysine ($K_{10}$) and poly-L-aspartic acid ($D_{10}$) form condensates at equimolar concentrations[75], exhibiting low interfacial tension and viscosity, see Supplementary Fig. 9 and Supplementary Table 1. Previous studies on $K_{10}/D_{10}$ condensates interacting with membranes showed that wetting transitions are achievable by adjusting membrane charge and salinity[12]. Again, for this system we incorporated 10 mol% of DOPS in the membrane to increase the condensate-membrane affinity. Figure 7c shows that, consistently with the other tested systems, increasing lipid packing reduced $K_{10}/D_{10}$ condensates wetting affinity.

Figure 7d shows the geometric factor and the intrinsic contact angle for PEG/dextran condensates in contact with GUVs, while Fig. 7e, f shows these parameters for the Syn1 and $K_{10}D_{10}$ condensates interacting with charged membranes. Across all systems, the data consistently align with the glycinin results: higher membrane packing decreases condensate affinity.

The tested condensates systems exhibit significant variability in material properties including viscosity, surface tension, hydrophobicity, and surface charge (summarized in Supplementary Table 1 and

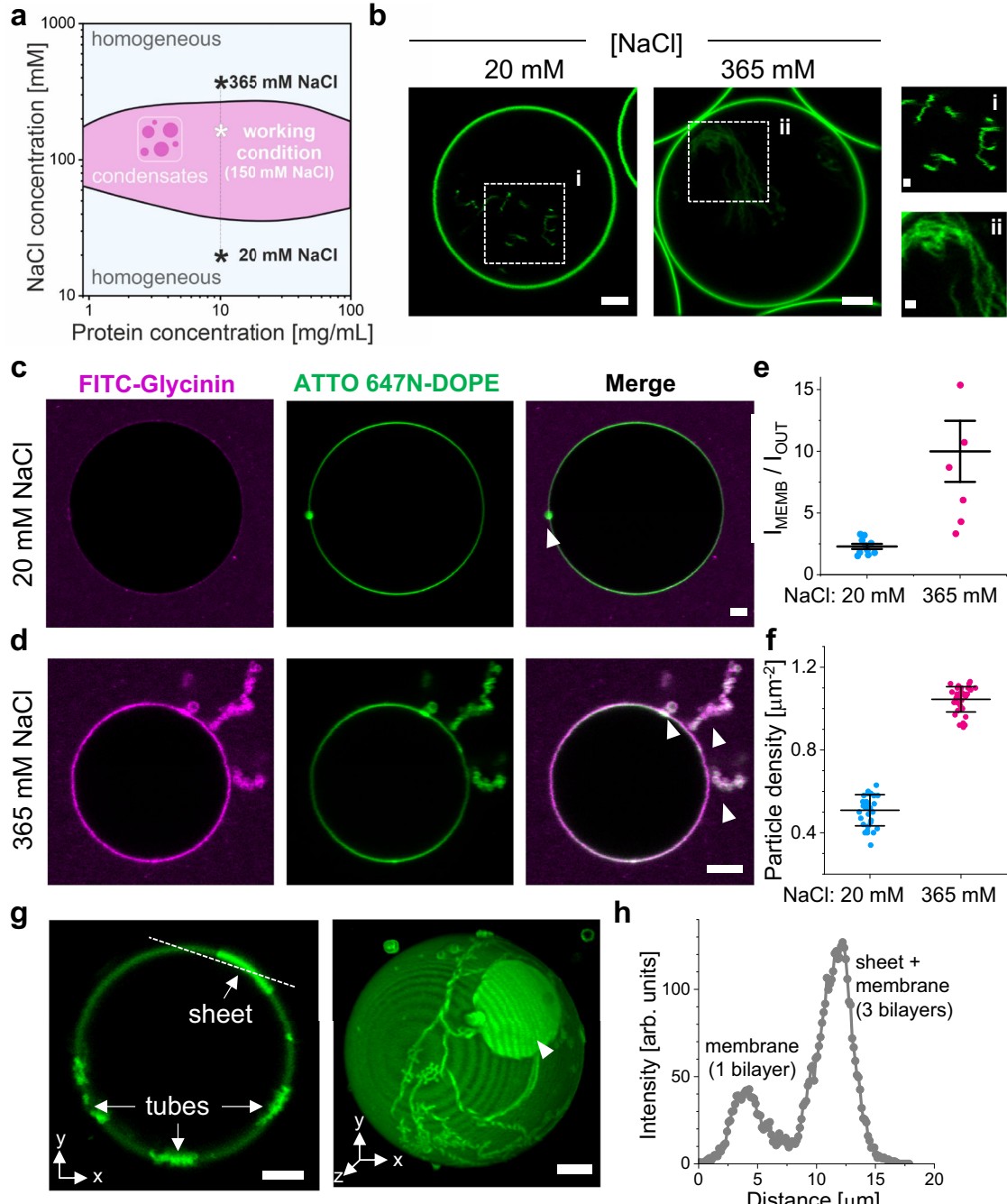

**Fig. 6 | Protein adsorption drives spontaneous formation of nanotubes and their transformation into double-membrane sheets at the vesicle surface.**
**a** Glycinin phase diagram as a function of NaCl concentration. The working condition for condensate formation, and two homogeneous solutions at low and high salinity are indicated. **b** DOPC GUVs grown in sucrose and then diluted in isotonic solutions of the indicated NaCl concentrations show inward tubulation due to the solution asymmetry. Scale bars: 5 μm, zoomed images: 1 μm. DOPC GUVs grown at 20 mM NaCl (**c**) or 365 mM NaCl (**d**) in contact with a homogeneous glycinin solution at the same NaCl concentration display outward bud and nanotube formation. Scale bars: 5 μm. **e** Ratio of the protein signal intensity (FITC-glycinin) at the membrane ($I_{MEMB}$) to the external solution ($I_{OUT}$), indicating protein binding to the membrane, which increases with higher salinity. Individual measurements are shown as dots and the lines indicate mean ± SD ($n = 10$). **f** Particle density

(reflecting the surface concentration of protein) obtained by mass photometry for 0.48 μg/mL glycinin solutions at the indicated NaCl concentration, over supported lipid bilayers of DOPC indicating higher adsorption with increasing salinity. Individual measurements are shown as dots and the lines indicate mean ± SD ($n = 35$). **g** Confocal microscopy cross-section (left) and 3D projection (right) of the membrane channel for a DOPC GUV in contact with a homogeneous glycinin solution at 365 mM NaCl. The tubes adhere to the vesicle surface and transform into double-membrane sheets; the double-membrane sheets essentially represent deflated pancake-like vesicles connected via a tube and adhering to the mother GUV. Scale bars: 5 μm. **h** Intensity profile across the dashed line shown in (**g**), indicating that the intensity for the double-membrane sheet adsorbed on the vesicle (3 bilayers) is three times higher than for the membrane (single bilayer). Source data are provided as a Source Data file.

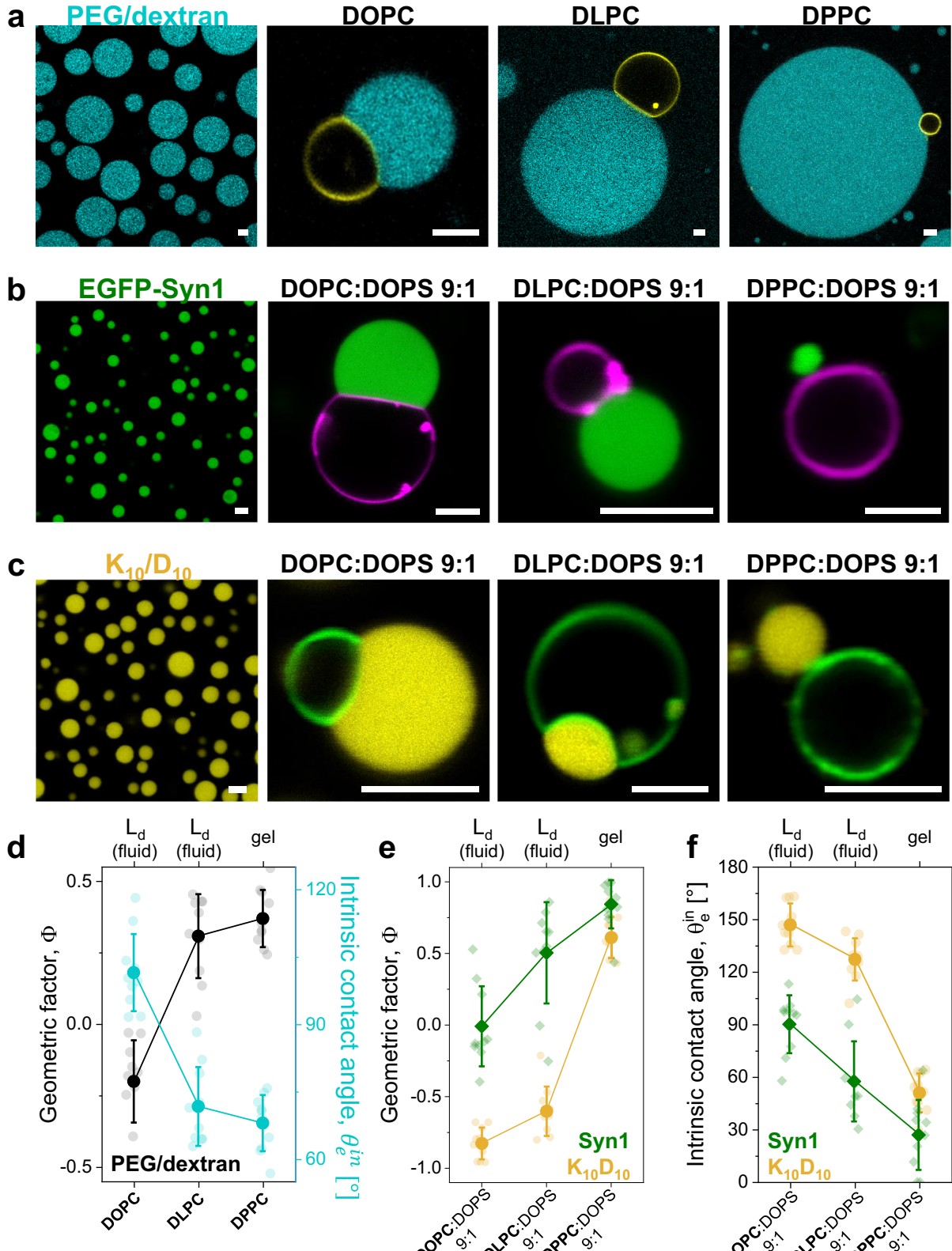

**Fig. 7 | Influence of lipid packing on condensate wetting is universal across different condensate systems. a–c** Confocal microscopy images of condensates isolated and in contact with GUVs labeled with 0.1 mol% ATTO 647N-DOPE for the indicated membrane compositions. **a** PEG/dextran labeled with 0.5% FITC-dextran (cyan) (**b**) EGFP-Synapsin 1. (**c**) $K_{10}/D_{10}$ labeled with 0.1 mol% of TAMRA-$K_{10}$ (yellow). **d** Geometric factor $\Phi$ (black circles), and intrinsic contact angle $\theta_e^{in}$ (cyan circles, right axis), for PEG/dextran condensates in contact with GUVs of the indicated compositions. Individual measurements are shown as dots and the lines indicate mean ± SD ($n$ = 10). **e** Geometric factor $\Phi$ for Syn1 (green diamonds) and $K_{10}/D_{10}$ condensates (yellow circles) in contact with vesicles of the indicated compositions. Individual measurements are shown as dots and the lines indicate mean ± SD ($n$ = 10). **f** Intrinsic contact angle $\theta_e^{in}$ for Syn1 (green diamonds) and $K_{10}/D_{10}$ condensates (yellow circles) in contact with membranes of the indicated compositions. Individual data points are shown for each membrane composition. The lines indicate the mean value ± SD ($n$ = 10). All scale bars: 5 μm. Source data are provided as a Source Data file.

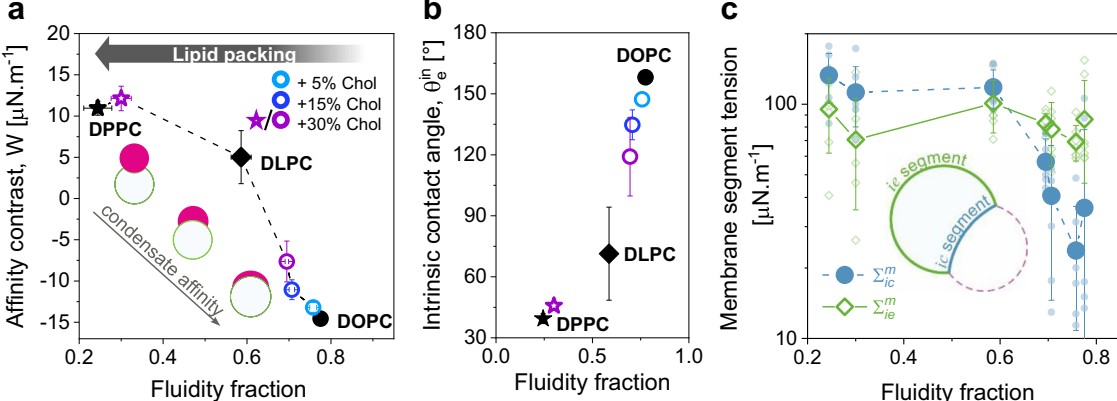

**Fig. 8 | Membrane fluidity modulates condensate affinity and membrane segment tension. a** Affinity contrast as a function of fluidity fraction for all tested membrane systems in contact with glycinin condensates. The symbols indicate mean ± SD with $n = 5$ for the fluidity fraction determination, and $n = 10$ for the affinity contrast. The dashed line is just a guide to the eye. The insets illustrate that condensate affinity for the membrane increases as $W$ decreases. **b** Intrinsic contact angle, $\theta_e^{in}$ for the same systems shown in (**a**). The symbols indicate mean ± SD with $n = 5$ for the fluidity fraction determination, and $n = 10$ for the intrinsic angle calculation. **c** Membrane tension for the *ie* and *ic* vesicle segments (respectively wetted by the protein-poor phase and by the condensate), as indicated in the sketch. Individual measurements are shown with small symbols and mean ± SD values are indicated with larger symbols ($n = 7$). Lines are a guide to the eye. All data were obtained for the previously defined working conditions. Source data are provided as a Source Data file.

Supplementary Figs. 10 and 11). ACDAN is a soluble analog of LAURDAN, which has been previously used to detect changes in the dipolar relaxation in condensates nano-environment using hyperspectral imaging[22,76]. By analyzing the spectral phasors of ACDAN, we compared the micropolarity of the different condensates, with pure water and ethanol serving as reference points (Supplementary Fig. 10). The condensates systems span over a wide range of dipolar relaxation values, with glycinin and Syn1 being the most hydrophobic, and PEG/dextran and $K_{10}/D_{10}$ exhibiting a more hydrophilic nano-environment. Additionally, we evaluated the surface charge of the protein/peptide-based condensates by measuring the ζ-potential using microelectroforesis[77]. We observed that despite that proteins and peptides can be highly charged in homogeneous aqueous solutions (e.g. glycinin ζ-potential ≈ −30 mV in water[78]), all condensates exhibit low ζ-potential (within a 0-13 mV range) under the experimental conditions (see Supplementary Fig. 11). This is likely due to the high ionic strength of the buffer solutions together with the screening that can occur due to protein reorganization during LLPS. Note that the PEG/dextran condensates are essentially neutral and thus this system has not been included in the analysis.

These results confirm that the observed correlation between wetting affinity and lipid packing is independent of condensate chemical and material properties, suggesting a general mechanism by which condensate-membrane interactions can be regulated by tuning lipid packing.

**Membrane fluidity correlates with affinity contrast and determines membrane tension**
Having established that condensate affinity for the membrane can be tuned through lipid packing, we examined the correlation between fluidity fraction and affinity contrast, $W$, across all tested membrane compositions (with and without cholesterol) in contact with glycinin condensates. Figure 8a demonstrates this direct correlation, which shows an almost linear trend when plotting the intrinsic contact angle against the fluidity fraction, as illustrated in Fig. 8b. This further confirms that lipid packing and hydration, rather than membrane phase state, primarily determine wetting interactions.

From the microscopic contact angles defined by the three interfaces shown in Fig. 2b and the condensate interfacial tension ($\Sigma_{ce}$), we calculated the tensions of the two membrane segments ($\Sigma_{ie}^m$, and $\Sigma_{ic}^m$)[45,48](see "Methods"). Figure 8c shows these tensions as a function of

fluidity fraction for all tested membrane compositions. While the tension for the membrane segment wetted by the condensate, $\Sigma_{ic}^m$, decreases with increased fluidity (the membrane segment becomes more floppy), the tension of the segment wetted by the external buffer, $\Sigma_{ie}^m$, remains approximately constant, regardless of lipid packing. This aligns with previous findings showing that high adhesion energy enables condensates to pull lipids together to the membrane-condensate interface[12,22]. It is important to emphasize that while the affinity contrast $W$ and the intrinsic contact angle $\theta_e^{in}$ are material properties, the calculated membrane tensions, $\Sigma_{ie}^m$ and $\Sigma_{ic}^m$, depend on the initial lateral stress of the GUVs, that can vary within the same sample, contributing to the observed spread of data in Fig. 8c.

## Discussion
Wetting of membranes by biomolecular condensates is a fundamental aspect of organelle interactions crucial to various cellular processes, involved in both physiology and disease[2,79]. The elucidation of these interactions has been greatly facilitated by in vitro systems, which allow precise control over physicochemical parameters and reduction of complexity compared to cellular environments[46]. Through such approaches, mechanisms underlying various membrane remodeling processes[12,13,18,69], coupling between membrane and protein phase separation[6,23,26,55], and impact of condensate wetting on membrane order and fluidity[12,22] have been uncovered, and even revealed cellular functions of condensates interacting with membranes[9].

In this study, we combined hyperspectral imaging and phasor analysis with estimates of fluid-elastic parameters from microscopy images, thereby assessing the wetting affinity of condensates for membranes as a function of lipid packing. Our results clearly demonstrate that the degree of lipid packing determines wetting affinity. Increasing lipid hydrocarbon chain length or saturation reduces condensate affinity for the membrane (Figs. 2 and 7). Additionally, we explored the effect of cholesterol, showing that higher cholesterol levels increase lipid packing and decrease condensate affinity (Fig. 3). Although we studied simple single-, two- and three-component model membranes, key material properties such as bending rigidity and lipid packing have been consistently reproduced in lipid-only membranes, effectively mimicking plasma membranes and extending the relevance of our results to biological systems[80].

Importantly, our findings show that condensate-membrane interactions are governed by lipid packing rather than membrane

phase state per se. For instance, membranes in a liquid-disordered phase state (e.g., DOPC and DLPC) exhibit different degrees of packing (Fig. 2), and this variability extends to cholesterol compositions (Fig. 3). This suggests that membrane wetting by condensates can be finely tuned through changes in membrane composition. It is important to emphasize that while the gel and $L_o$ phases with different fluidities demonstrate low affinity or dewetting under the working conditions and across condensate systems (Supplementary Figs. 1–3), this does not imply that condensates cannot wet these phases. Rather, it suggests that significantly higher condensate-membrane affinity is required to observe wetting by these phases. For example, the PEG/dextran condensates, display a Φ value of 0.4 for the gel phase, indicating higher affinity compared to the other tested systems (Fig. 7).

The approach used in this work, namely combining hyperspectral imaging of LAURDAN to build a fluidity scale and determining fluid-elastic parameters from the microscopy images, allowed us to determine the tensions of the wetted and bare membrane segments for different membrane systems in contact with glycinin condensates (Fig. 8c). This information, which is difficult to obtain by other experimental methods, further confirms our previous studies showing that at higher condensate-membrane affinities the lipids at the condensate-membrane interface are pulled together triggering interfacial ruffling when there is enough excess membrane[12,22].

Previously, we have shown that condensates wetting influences lipid packing and hydration[22]. Moreover, molecular dynamic simulations[81], and FRAP measurements which demonstrate a decrease in diffusion coefficients at the membrane-condensate interface for various condensate systems[12,17], strongly support the idea that this is a general mechanism of membrane-condensate interaction. In this study, we further reveal that the initial state of lipid packing, in turn, regulates condensate affinity for the membrane. This finding is validated for a variety of condensate systems, suggesting this regulatory mechanism is a universal phenomenon (Figs. 2, 7). The regulatory mechanism is supported by evidence showing that condensate affinity increases with photo-induced membrane area expansion, which reduces packing[82].

These results underscore the crucial role of the lipid interface in mediating the interaction. Considering that the water activity at the interface decreases with increasing lipid packing[29], the dynamics of the interfacial water most likely influences the condensate-membrane interaction. One plausible mechanism is that the interaction between the condensate and the membrane requires dehydration of the interface. In other words, condensates exhibit a preference for well-hydrated membranes. This would explain why tightly packed membranes, which are already dehydrated, show reduced affinity for condensates compared to loosely packed, highly hydrated membranes. Then, upon interaction, the condensate-membrane affinity drives a localized increase in lipid packing[22]. In this sense, the physical state of water has been shown to provide a link between protein structure in bulk and structural changes in lipid membranes[28]. Moreover, cholesterol addition alters the alignment of interfacial water and the membrane dipole potential[83], potentially facilitating the specific association of condensates with cellular organelles of varying cholesterol content[84].

By tethering proteins to the membrane with specific anchors (e.g. NTA lipids, PEGylated or cholesterol-based lipids linked to poly-uridine), it is possible to enhance condensate interaction with specific membrane lipid phases[6,23,24,85]. Our results using non-tethered 3D condensates reveal that lipid packing alone, in the absence of specific protein-lipid interactions, dictates condensate specificity for a particular lipid phase (Fig. 4).

Electrostatics often dominates membrane-condensate interactions, with lipid charges playing a regulatory role in membrane-condensate affinity[12,19,86]. Here, charged lipids were employed to increase the initial condensate-membrane affinity for Syn1 and $K_{10}/D_{10}$

systems. As shown in Supplementary Fig. 7, membranes composed of DOPC, DLPC, and DPPC, each containing 10% DOPS, exhibit differences in lipid packing density which are ordered according to DPPC:DOPS > DLPC:DOPS > DOPC:DOPS. Based on this trend, one might expect that increased lipid packing, while maintaining the same fraction of charged lipids, would raise the charge density and thereby strengthen the condensate-membrane affinity. However, the opposite effect was observed: membranes with higher packing density exhibited reduced condensate-membrane affinity. This effect, which is consistent with observations for neutral membranes, indicates that increased lipid packing weakens affinity even in systems where electrostatics favors the interactions. These findings, validated for two very different systems, Syn1 and $K_{10}/D_{10}$, clearly point to the role of lipid packing as a modulator of condensate-membrane interaction, extending beyond purely electrostatic interactions.

While in this work we focused on 3D non-anchored condensates, evaluating the impact of lipid packing in systems with specific lipid-protein interactions could provide additional insight into the similarities and differences in interaction mechanisms. When 2D condensates form at the membrane surface via protein binding to NTA lipids, the condensate-membrane affinity can be regulated by varying the concentration of NTA lipids[13,87]. However, when specific protein-lipid interactions drive membrane wetting, predicting the effect of lipid packing might be challenging, since lipid sorting could arise upon condensate interaction. Moreover, in the case of NTA mediated protein binding, fluorescence quenching by nickel[88] complicates the use of fluorescence-based techniques, such as those employed in this work, by affecting the dye lifetime and quantum yield. Alternative systems, such as the specific interaction between the epsin1 N-terminal homology (ENTH) domain and PI(4,5)P2 lipids[17], could provide a suitable approach to address this issue. Exploring the effect of lipid packing in systems with specific protein-lipid interactions is beyond the scope of this work. Nonetheless, it is important to note that NTA-lipids, often used to investigate tethered condensates, are synthetic and not naturally present in biological membranes. This further underscores the value of studying non-tethered or naturally tethered (e.g. via PIP lipids or GPI anchors) condensates to gain insights into physiologically relevant interactions with natural membranes.

Condensates are capable of inducing extensive membrane remodeling, including interfacial ruffling, tube formation and double-membrane sheet generation[13,46,67,69,89]. Here, we observed that protein adhesion promotes the formation of tubular structures at the condensate-membrane interface (Fig. 5), facilitated by spontaneous curvature generation (Fig. 6). Notably, protein adsorption also drives the formation of double-membrane sheets (Fig. 6g-h, S6, Supplementary Movie 7), reminiscent of processes observed in organelle morphogenesis, such as that of autophagosomes[90] and the endoplasmic reticulum network of interconnected membrane tubes and sheets[91]. The formation of double-membrane sheets can be attributed to the significant excess area present in the vesicles. Storing this excess area in double-membrane sheets is more efficient than storing it in nanotubes (considering their different area-to-volume ratios). The adhesion of these sheets to the vesicle membrane is mediated by proteins, and the increased local protein concentration, arising from proteins adsorbed on both the sheet and the GUV membranes, could potentially lead to 2D phase separation. This process may result in the formation of flat, two-dimensional condensates, similar to those described in previous studies[6,13,92].

In summary, we have unveiled a regulatory mechanism by which condensate wetting is modulated, allowing specificity for a distinct lipid phase. Both lipid chain length and cholesterol content can influence wetting and membrane remodeling (Fig. 9). While the dehydration of the interface is a plausible mechanism that could explain the observed behavior, the question of what drives protein binding to the membrane in the absence of lipid anchors or tethers, as in the systems

presented here, remains and would require further investigation. Atomistic and coarse-grained simulations suggest that there is no intercalation of condensate molecules in the membrane and that electrostatic interactions play an important role[81,93,94], even in the absence of charged headgroups[95]. Thus, assessing the electrical properties of condensates is crucial to unraveling their interaction mechanism. The study of membrane-condensate interfaces is challenging, but key to understanding the wetting and remodeling processes orchestrated by condensates.

## Methods
### Materials
The lipids 1,2-dioleoyl-sn-glycero-3-phosphocholine (DOPC), 1,2-dilauroyl-sn-glycero-3-phosphocholine (DLPC), 1,2-dipalmitoyl-sn-glycero-3-phosphocholine (DPPC), 1,2-dioleoyl-sn-glycero-3-phospho-L-serine (DOPS), Sphingomyelin from chicken egg (SM), and cholesterol, were purchased from Avanti Polar Lipids (IL, USA). The fluorescent dye 6-dodecanoyl-2-dimethylaminonaphthalene (LAURDAN) was purchased from Thermofisher Scientific (USA). ATTO 647N-DOPE was obtained from ATTO-TEC GmbH (Siegen, Germany). 2-Acetyl-6-(dimethylamino) naphthalene (ACDAN) was obtained from Santa Cruz Biotechnology (USA). Chloroform obtained from Merck (Darmstadt, Germany) was of HPLC grade (99.8 %). The lipid stocks were mixed as chloroform solutions at 4 mM, containing 0.1 mol% ATTO 647N-DOPE or 0.5 mol% LAURDAN, and were stored until use at −20 °C. Fluorescein isothiocyanate isomer (FITC), bovine serum albumin (BSA, fatty acid free), sucrose, glucose, dimethyl sulfoxide (DMSO), hydrochloridric acid (HCl), sodium hydroxide (NaOH), sodium bisulfite, sodium chloride (NaCl), potassium chloride (KCl), magnesium chloride (MgCl$_2$), Tris HCl buffer (pH=7.4), Tris(2-carboxyethyl)phosphin-hydrochlorid (TCEP), dextran from Leuconostoc spp (Mw 450−650 kg/mol), fluorescein isothiocyanate-dextran (Mw 500 kg mol$^{-1}$), poly(ethylene glycol) (PEG 8 K, Mw 8 kg/mol), ethanol absolute (99.5%), and Polyvinyl alcohol (PVA, Mw 145000), were obtained from Sigma-Aldrich (Missouri, USA). The oligopeptides, poly-L-lysine hydrochloride (degree of polymerization, $n = 10$; K$_{10}$) and poly-L-aspartic acid sodium salt (degree of polymerization, $n = 10$; D$_{10}$) were purchased from Alamanda Polymers (AL, USA) and used without further purification (purity≥95%). A N-terminal TAMRA-labeled K$_{10}$ was purchased from Biomatik (Ontario, Canada). All aqueous solutions were prepared using ultrapure water from a SG water purification system (Ultrapure Integra UV plus, SG Wasseraufbereitung) with a resistivity of 18.2 MΩ cm.

### Giant vesicle preparation
Giant unilamellar vesicles were prepared by the electroformation method[96], except where indicated. Briefly, 3 μL of the desired lipid solution were spread onto indium tin oxide (ITO)-coated glasses and dried under vacuum for 1 h. A chamber was assembled using a Teflon spacer and filled with 1.9 mL of the swelling solution. Then, a sinusoidal electric field of 1.0 Vpp and 10 Hz was applied using a function generator for 1 h. For the experiments with condensates, a sucrose solution was used for swelling. In all cases, the solution osmolarities were carefully adjusted using a freezing-point osmometer (Osmomat 3000, Gonotec, Germany).

The GUVs for the experiments in Fig. 6, were prepared with the PVA gel-assisted swelling method[97], allowing vesicle swelling in high salinity conditions. Briefly, two coverslips were cleaned with water and ethanol and dried under nitrogen. A 40 mg/mL PVA solution was prepared by heating at 90 °C while stirring for 3 h. A 20 μL aliquot of the PVA solution was spread on the glass slides and dried for 1 h at 60 °C. A 3-4 μL layer of lipid stock solution was deposited on the PVA-coated glass and kept for 1 h under vacuum at room temperature. The chamber was assembled with a 2 mm-thick Teflon spacer and filled with 1 mL of the desired NaCl solution. After 30 min, the vesicles were carefully harvested in order to prevent PVA detachment from the cover glass.

When using different solutions for the vesicle growth and condensate formation, the osmolarities were always matched between the suspensions before mixing. In general, to promote condensate interaction, vesicles should possess some excess membrane area to allow deformation. GUVs samples are typically heterogeneous in terms of membrane tension, different vesicles exhibiting varying amounts of excess membrane. This excess area can be increased by vesicle deflation, which can be achieved by slightly increasing the osmolarity of the external solution (e.g. by approximately 5-10% compared to the internal solution) before mixing.

### Preparation of small unilamellar vesicles
Small unilamellar vesicles (SUVs) of pure DOPC were prepared at a total lipid concentration of 500 μM and used for the preparation of supported lipid bilayers. To prepare the SUVs, lipids were dried under vacuum for at least 2 h at room temperature, then resuspended in 1 mL of buffer (20 mM Hepes, 150 mM KCl, pH 7.4). The glass vial was covered with Parafilm, incubated at 42 °C for 30 min, vortexed and the content transferred to a 1.5 mL Eppendorf tube. Sonication was performed using a 2 mm tip (Sonopuls MS 72, Bandelin) for 30 min total time, (5% cycle, 20% amplitude) on an ice bath. The resulting suspension was centrifuged at 21,000 × g for 30 min, and the supernatant containing SUVs was collected.

### Supported lipid bilayer (SLB) formation for mass photometry
Coverslips (24 × 50 mm, Menzel Gläser) were cleaned by alternating spraying isopropanol and Milli Q water for 3x and dried using compressed air. Coverslips were then treated with UV/Ozone (UV/Ozone ProCleaner™, Bio Force Nanosciences) for 20 min.

To form supported lipid bilayers (SLBs) a Silicon gasket (CultureWell™ CW-8R-1.0- Gasket, 8−6 mm diameter x 1 mm depth, 15−30 μL, Grace Bio-Labs) was placed on a cleaned glass coverslip. 30 μL of SLB buffer (20 mM Hepes, 150 mM KCl, 1.7 mM MgCl2, pH 7.4) was added, followed by 20 μL of SUVs. The mixture was incubated for at least 20 min in a home-build humidity chamber. After incubation, the SLB was washed extensively with SLB buffer and the buffer was exchanged to either 20 mM or 365 mM NaCl, adjusting the final volume in the well to 60 μL.

A 6.8 μg/mL glycinin stock solution was prepared in either 20 mM or 365 mM NaCl. 4.5 μL of this stock solution was added to the well (to a final concentration of glycinin of 0.48 μg/mL) and incubated for 5 min before data acquisition.

### Protein expression, extraction, purification, and labeling
**Glycinin**. Glycinin was purified as described by Chen et al. [47]. Initially, the flour was mixed with 15 times its weight in water, and the pH was adjusted to 7.5 using sodium hydroxide. After separating the insoluble material by centrifugation (30 min, 9000 × g, 4 °C), sodium bisulfite was added to the resulting supernatant to a final concentration of 0.98 g/L. The pH was then lowered to 6.4 with hydrochloric acid, and the solution was allowed to stand overnight at 4 °C. After centrifugation (30 min, 6500 × g, 4 °C), the glycinin-enriched precipitate[47] was collected and redissolved in water (fivefold), adjusted to pH 7, and extensively dialyzed against pure water at 4 °C. Finally, the glycinin solution was freeze-dried, yielding a product with 97.5% purity, confirmed by SDS-PAGE[47].

To fluorescently label the glycinin, a 20 mg/mL solution was prepared in 0.1 M carbonate buffer at pH 9. A solution of FITC in DMSO was gradually added to the protein solution, reaching a final FITC concentration of 0.2 mg/mL. The solution was incubated in the dark at room temperature for three hours. Unbound FITC was removed using a Sephadex G-25 desalting column (GE Healthcare, IL, USA), and the buffer was exchanged with ultrapure water. The pH of the labeled glycinin was adjusted to 7.4 with sodium hydroxide. For fluorescence microscopy, a 4%v/v of the labeled glycinin was added to unlabeled glycinin.

## a  Higher affinity for less packed membranes

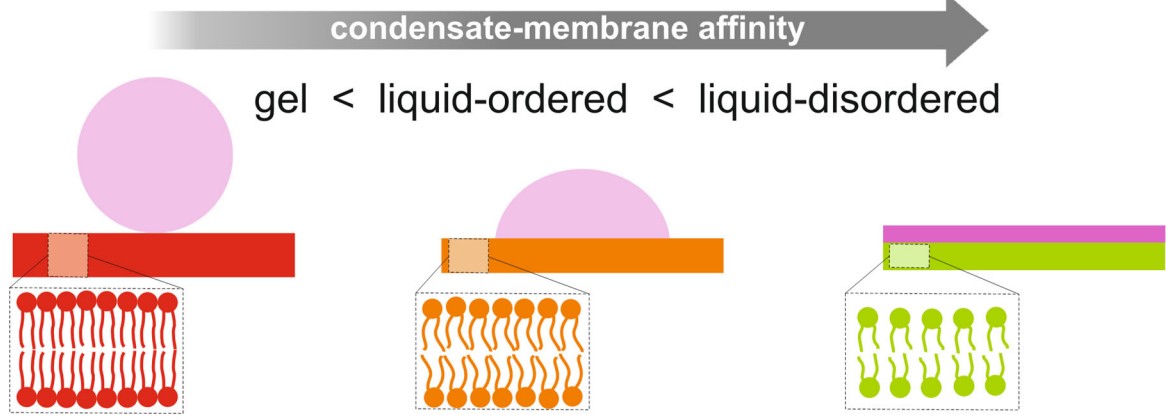

## b  Wetting selectivity for less packed domains

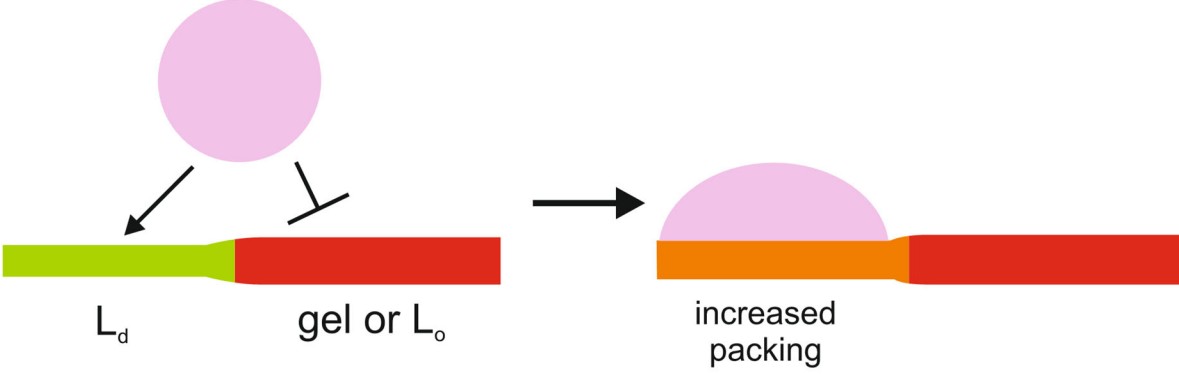

## c  Morphological transformations and curvature generation

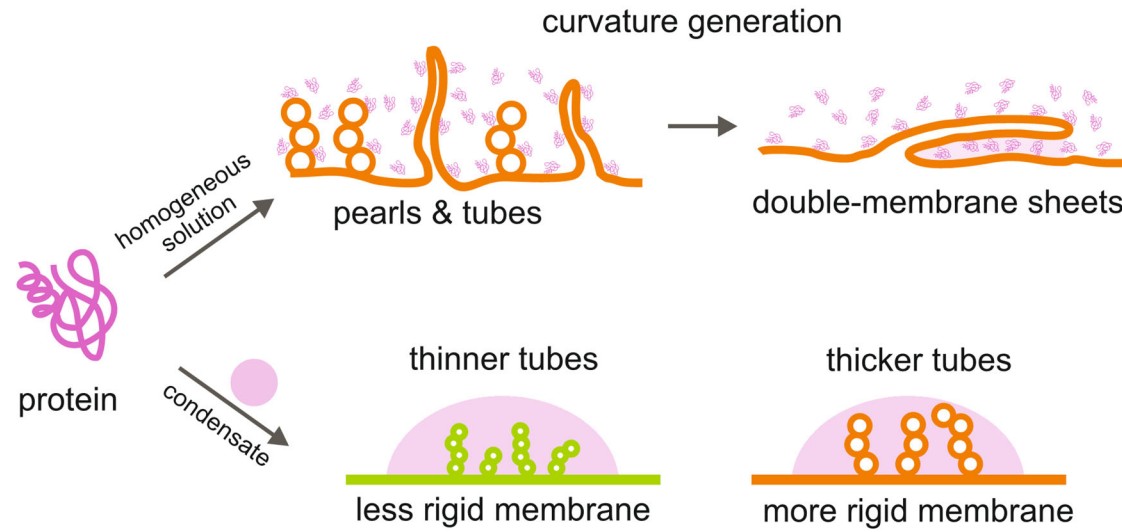

**Fig. 9 | Sketch summarizing the main findings. a** The wetting affinity of biomolecular condensates is higher for less densely packed membranes, and can be tuned by changing the lipid chain length, the degree of chain saturation, or the cholesterol content. **b** When in contact with phase-separated membranes, condensates preferentially interact with the less densely packed domains, locally increasing the lipid packing[22]. **c** Protein adsorption from homogeneous protein solutions can induce membrane spontaneous curvature, triggering the formation of necklace-like pearls and tubes. When substantial excess area is available, these structures may interconvert into double-membrane sheets, which adhere to the vesicle surface mediated by proteins (top). Upon interaction with vesicle membranes with excess area, condensates can induce tubulation with tube size depending on the rigidity of the membrane (bottom).

## Synapsin 1 (Syn1) expression and purification

EGFP-tagged Synapsin 1 was purified as described by Hoffmann et al.[98]. In brief, protein was expressed in Expi293F™ cells (Thermo Fisher Scientific) for three days post enhancement following manufacturer guidelines. Harvested cells were lysed by three freeze-thawing cycles in a buffer containing 25 mM Tris-HCl (pH 7.4), 300 mM NaCl, 0.5 mM TCEP (buffer A), and Roche cOmplete EDTA-free protease inhibitors. All following purification steps were performed at 4 °C. The lysate was cleared by centrifugation (1 h at 30,000 × g) and subjected to immobilized metal affinity chromatography (IMAC) using a Ni-NTA column (HisTrap™ HP, Cytiva) in buffer A with varying imidazole concentrations (25 mM imidazole for binding, 40 mM for washing and 400 mM for elution, respectively). Eluates were concentrated (Amicon® Millipore Centrifugal Filters) and subjected to size exclusion chromatography (Superdex™ 200 Increase 10/300, Cytiva) in 25 mM Tris-HCl (pH 7.4), 150 mM NaCl, 0.5 mM TCEP.

For untagged Synapsin 1 purification, Synapsin 1 was expressed as His-SUMO-tag fusion in Expi293F™ cells with subsequent removal of the His-SUMO tag during the purification procedure by SENP protease as described by Hoffmann et al.[73]. For batch IMAC purification, cleared supernatant after cell lysis was incubated with cOmplete™ His-tag purification resin under constant agitation at 4 °C for 1 h. Washing steps (buffer A with 15 mM imidazole) and elution (buffer A with 400 mM imidazole) were carried out in a polyprep column (Biorad). Eluates were concentrated (Amicon® Millipore Centrifugal Filters) and subjected to size exclusion chromatography (Superdex™ 200 Increase 10/300, Cytiva) in buffer A. For overnight His-SUMO-tag cleavage, elution fractions containing His-SUMO-Synapsin 1 were combined and supplemented with SENP_EuB protease (protease:protein ratio of 1:20). Tag-removal was performed by reverse batch IMAC in buffer A supplemented with 15 mM imidazole. Tag-free Synapsin 1 was subjected to buffer exchange (25 mM Tris-HCl (pH 7.4), 150 mM NaCl, 0.5 mM TCEP) using a PD-10 column (Cytiva) and concentrated using a 30 K MWCO Amicon® Millipore Centrifugal Filter.

The purity of protein was validated by SDS-PAGE electrophoresis, similarly as in ref. 73. Proteins were snap-frozen in liquid nitrogen and stored at -80°C until further use.

## Formation of Syn1 condensates and interaction with GUVs

Condensates were formed as previously reported[73]. Briefly, the protein was mixed with buffer to a final concentration of 5 µM. The buffer consisted of 25 mM Tris, 150 mM NaCl, and 0.5 mM TCEP. An aliquot of PEG 8 K was added to a final concentration of 3% to trigger phase separation of the EGFP-Syn1 condensates, and 5% for the untagged protein. After 15 min of condensate growth and coalescence, a small aliquot (2%) of vesicles of the desired composition grown in an isosmotic sucrose solution were added.

## Preparation of glycinin condensates and glycinin-GUVs suspensions

A stock solution of glycinin was made by dissolving the protein in ultrapure water to a concentration of 20 mg/mL (pH=7), following by filtration with a 0.45 µm filter. For condensate formation, the stock solution was combined with an equal volume of a sodium chloride (NaCl) solution prepared at twice the desired final concentration (10 mg/mL final protein concentration)[12,47]. Prior to mixing with vesicles, the condensate suspension was diluted fourfold in NaCl.

Separately, vesicles were diluted tenfold into a NaCl solution matching the final NaCl concentration of the condensate dispersion. Condensates were mixed with the vesicle suspension at a 15% v/v, resulting in a final condensate concentration of 0.4 mg/mL. Glass coverslips (26×56 mm, Waldemar Knittel Glasbearbeitungs GmbH, Germany) were cleaned with EtOH and water before passivation with a 2.5 mg/mL BSA solution. For microscopy observation, a chamber was assembled using a round silicone spacer. After addition of an aliquot of the vesicle-condensate suspension the chamber was closed with another coverslip.

## Preparation of oligopeptides $K_{10}/D_{10}$ coacervates and $K_{10}/D_{10}$-GUVs suspensions

Phase separation was triggered by gently mixing aliquots of stock solutions of KCl, MgCl$_2$, glucose, D$_{10}$ and K$_{10}$ (in this order) to a final volume of 20 µL. For labeling, a 0.1 mol% solution of TAMRA-K$_{10}$ in water was added. The final concentration of each component was: 15 mM KCl, 0.5 mM MgCl$_2$, 170 mM glucose, 2 mM D$_{10}$, and 2 mM K$_{10}$. The final osmolality of the mixture was ≈200 mOsm/kg.

For the interaction of membranes with $K_{10}/D_{10}$ condensates, the vesicle suspension was diluted 1:10 in the final buffer of the corresponding droplet suspension. An aliquot of this diluted vesicle solution was then mixed with the droplet suspension in an 8:1 volume ratio directly on the cover glass and sealed for immediate observation under the microscope.

## PEG/dextran condensates in contact with GUVs

Phase separation of the PEG/dextran solution was achieved by mixing the polymers in weight fractions 6.42%:4.09% in deionized water, which corresponds to a (1:1.57) molar ratio of PEG:dextran[49]. A 0.5% of FITC-labeled dextran was included to observe the dextran-rich condensates. After bulk phase separation was observed, an aliquot of the PEG-rich phase was placed on the microscope slide and vesicles grown in the same PEG-rich phase were added to observe the interaction.

## ACDAN labeled condensates

For the experiments shown in Supplementary Fig. 10, the phase separation was triggered for the unlabeled protein/polymer/peptide condensates in the presence of 5 µM ACDAN.

## Hyperspectral imaging

Hyperspectral images were acquired using the xyλ mode of a Leica SP8 FALCON confocal microscope using a 63 × 1.2 NA water immersion objective (Leica, Mannheim, Germany). The image acquisition was performed in the range 416–728 nm divided on 32 channels with a bandwidth of 9.75 nm. The excitation source was a pulsed Ti:Sapphire laser MaiTai (SpectraPhysics, USA), with a repetition rate of 80 MHz. Two-photon excitation was achieved at 780 nm for LAURDAN and ACDAN. The image size was 512 × 512 pixels$^2$ with a pixel size of 72 nm. The hyperspectral data were processed using the SimFCS software developed by the Laboratory of Fluorescence Dynamics, available at https://www.lfd.uci.edu/globals/.

## Spectral phasor plot

The phasor transform was used to analyze the hyperspectral data for LAURDAN and ACDAN. This allows to obtain the real and imaginary components of the Fourier transform namely G and S, respectively. The expressions below define the cartesian coordinates (G,S) of the spectral phasor plot[21]:

$$G = \frac{\int_{\lambda_{min}}^{\lambda_{max}} I(\lambda) \cos\left(\frac{2\pi n(\lambda - \lambda_i)}{\lambda_{max} - \lambda_{min}}\right) d\lambda}{\int_{\lambda_{min}}^{\lambda_{max}} I(\lambda) d\lambda} \tag{1}$$

$$S = \frac{\int_{\lambda_{min}}^{\lambda_{max}} I(\lambda) \sin\left(\frac{2\pi n(\lambda - \lambda_i)}{\lambda_{max} - \lambda_{min}}\right) d\lambda}{\int_{\lambda_{min}}^{\lambda_{max}} I(\lambda) d\lambda} \tag{2}$$

where for a given pixel $I(\lambda)$ is the intensity as a function of wavelength, measured between $(\lambda_{min}; \lambda_{max})$. The harmonic, $n$, represents the number of cycles of the trigonometric function fit in the wavelength range ($n$=1 for this work).

The phasor position encodes information about the spectral center of mass which is related to the angle, and the spectrum broadness is related to the distance from the plot center.

The linear combination rules of phasors[44] imply that when two independent fluorescent species as present in the sample, they fall in the phasor plot in a position resulting from the linear combination of the positions of the two "pure" independent species. Then, the fraction of each component can be determined by the coefficients of the linear combination.

## Two-component analysis

To analyze variations in dipolar relaxation sensed by LAURDAN or ACDAN, we employed the two-component (or two-cursor) approach. This analysis leverages the linear combination properties of the phasor plot[44] producing pixel distribution histograms along the linear trajectory (as shown in Fig. 1b). The histograms are the normalized number of pixels at each step along the trajectory between two cursors. For each histogram, we plotted the average value ± standard deviation. To be able to perform quantitative analysis with descriptive statistics, we calculated the center of mass of the histogram as follows:

$$CM = \frac{\sum_{i=0}^{i=1} F_i \, i}{\sum_{i=0}^{i=1} F_i} \qquad (3)$$

where $F_i$ is the fraction for fluidity or dipolar relaxation, respectively for LAURDAN or ACDAN experiments. Note that despite the cursor positions can be arbitrarily determined, the existence of any differences between the center of mass of the histograms are established through statistical analysis.

It is important to remark that in this work we define fluidity as any changes occurring in lipid rotational or translational rates at the headgroup-chain interface[99].

## Contact angles measurement and geometric factor calculation

To measure the apparent contact angles from the confocal microscopy images, we first determine the correct projection from z-stacks by aligning the rotational axis of symmetry of the GUV and the condensate. Otherwise, an incorrect projection will lead to a misleading interpretation of the system geometry and incorrect contact angles. Then, by considering that the vesicle, the droplet, and the vesicle-droplet interface correspond to spherical caps, we fit circles to their contours to extract the corresponding angles from geometry[12]. A detailed explanation of the contact angle measurement and the fluid-elastic parameters used in this work has been published elsewhere[11,12,45]. Briefly, the tension triangle in Fig. 2b implies the relationships[11]:

$$\frac{\Sigma_{ie}^m}{\Sigma_{ce}} = \frac{\Sigma + W_{ie}}{\Sigma_{ce}} = \frac{\sin \theta_c}{\sin \theta_i} \quad \text{and} \quad \frac{\Sigma_{ic}^m}{\Sigma_{ce}} = \frac{\Sigma + W_{ic}}{\Sigma_{ce}} = \frac{\sin \theta_e}{\sin \theta_i} \qquad (4)$$

between the surface tensions and the contact angles, as follows from the law of sines. Here, $W_{ic}$ and $W_{ie}$ are the respective adhesion parameters of the ic and ie membrane segments respectively in contact with the condensate and the external buffer (Fig. 2b). From the measured contact angles $\theta_e$, $\theta_i$, $\theta_c$, and the condensate surface tension, $\Sigma_{ce}$, it is possible to calculate the tensions of the membrane segments $\Sigma_{ic}^m$ and $\Sigma_{ie}^m$, as shown in Fig. 8b. The affinity contrast, $W$, between the condensate and the external buffer is given by:

$$W \equiv W_{ic} - W_{ie} = \Sigma_{ic}^m - \Sigma_{ie}^m \quad \text{with} \quad -\Sigma_{ce} \le W \le +\Sigma_{ce} \qquad (5)$$

The limiting value $W = -\Sigma_{ce}$ corresponds to complete wetting by the condensate phase whereas the limiting case $W = +\Sigma_{ce}$ describes dewetting from the condensate phase. When taking the difference between the two equations in 4, the affinity contrast, W, becomes:

$$W = \Phi \Sigma_{ce} \quad \text{with} \quad \Phi \equiv \frac{\sin \theta_e - \sin \theta_c}{\sin \theta_i} \qquad (6)$$

The rescaled affinity contrast, $W/\Sigma_{ce}$, is a mechanical quantity related to the adhesion free energies of the membrane segments, and is equal to the geometric factor, $\Phi$, that can be obtained from the three contact angles. The inequalities in Eq. (5) imply $-1 \le \Phi \le 1$ for the geometric factor, $\Phi$. When $\Phi = -1$ there is complete wetting of the membrane by the condensate phase, while $\Phi = +1$ corresponds to dewetting of the membrane by this phase. The dimensionless factor, $\Phi$ is negative if the membrane prefers the condensate over the exterior buffer and positive otherwise. Note that $\Phi$ is scale-invariant and does not depend on the relative sizes of a given vesicle-condensate couple[12].

At the nanoscale, the condensate-membrane affinity is defined by the intrinsic contact angle[50]. Here, we consider the intrinsic contact angle that opens towards the external solution, $\theta_e^{in}$ (as shown in Fig. 2b), that relates to the geometric factor through:

$$\cos \theta_e^{in} = (\sin \theta_e - \sin \theta_c)/\sin \theta_i \qquad (7)$$

## STED microscopy

To obtain the super-resolution images, an Abberior STED setup (Abberior Instruments GmbH) mounted on an Olympus IX83 microscope (Olympus Inc., Japan) equipped with a 60×,1.2 NA water immersion objective was used. For fluorescence excitation and depletion 640 nm and 775 nm pulsed laser were used, respectively. The alignment was performed using 150 nm gold beads (Sigma-Aldrich, USA), and 100 nm TetraSpeck™ beads (Invitrogen, USA) where used to correct any mismatches between the fluorescence and scattering modes. The resolving power of the setup was ~35 nm was at 80% STED laser power, tested on 26 nm crimson beads (FluoSpheres™, Molecular Probe)[49]. For our experiments, we used 3D STED (instead of 2D STED), since it allows to eliminate the out-of-focus signal. The pixel size was 50 nm with a dwell time of 10 μs.

## Fluctuation spectroscopy

To measure the bending rigidity of GUVs composed of pure DOPC or the binary mixture DOPC:DPPC 9:1, fluctuation analysis was performed. For that purpose, GUVs were grown in sucrose by electroformation and diluted tenfold in a glucose solution slightly hypertonic (~5 %) to deflate the GUVs. Then GUVs were visualized under phase contrast using a ×40 objective on a Zeiss AXIO Observer D1 microscope. Image sequences of 3000 frames were taken using a pco.edge sCMOS camera (Excelitas Technologies, Waltham, MA, USA) at a rate of 25 frames per second (fps) with 200 μs exposure. The bending rigidity was obtained by the Fourier decomposition of thermally driven membrane fluctuations into spherical modes[53].

## FRAP measurements

For FRAP measurements the Leica SP8 setup was used. A 2 μm circular region of interest (ROI) was used and condensates were bleached during ~3 s. FRAP curves were build using ImageJ.

## Mass photometry data acquisition and analysis

Mass photometry data were acquired using a OneMP instrument (Refeyn Ltd) on a detection area of 10.8 μm × 6.8 μm, at 270 Hz for 30 s with frame binning set to 2. Data analysis followed the procedure described by Foley et al. [100], using the Python scripts provided by the authors with minor adjustments according to the device specifications. Mass calibration was performed in the absence SLB. The particle density was obtained by averaging counted particles per area in each frame.

## Microelectrophoresis

The $\zeta$-potential of protein/peptide-based condensates were measured based on their electrophoretic mobility according to the method introduced in Van Haren et al.[77]. Briefly, condensates were placed on a glass slide to which thin copper electrodes were fixed. The condensates were exposed to a direct current (DC) field by connecting an Agilent 33220 A function generator to the copper electrodes. The electric field was applied for 1000 s and the voltage varied from 2-10 V depending on the desired field intensity (see Supplementary Fig. 11). The electrophoretic motion of condensates induced by the electric field was recorded under bright-field confocal microscopy and the drift velocity $v$ was computed based on the projected trajectory of condensates along the axis parallel to the direction of the electric field. Values of the $\zeta$-potential were determined using a modified form of the well-known Smoluchowski equation that accounts for the liquid properties of condensates[77]:

$$\zeta = \frac{3\eta_c \nu}{\epsilon_0 \epsilon_r E}\left(\frac{1}{3\eta_e + \kappa R}\right), \qquad (8)$$

where $\eta_c, \eta_e, \kappa, R, \epsilon_0, \epsilon_r$ and $E$ are respectively the condensate viscosity, external solution viscosity, inverse Debye length, condensate radius, permittivity of empty space, relative permittivity of the external solution, and the norm of the electric field.

## Statistics and reproducibility

At least three independent experiments were used to perform the statistical analysis. Pixel histograms are shown as means ± standard deviation (SD). The center of mass measurements are represented as scatter plots containing the individual measurements and the mean values ± SD. Results were analyzed using One-way ANOVA and Tukey post-test analysis ($p < 0.0001$, **** | $p < 0.001$, *** | $p < 0.01$, ** | $p < 0.05$, * | ns = non-significant). Statistical analyses and data processing were performed with the Origin Pro software (Originlab corporation). All the microscopy images shown are representative of at least three independent experiments.

## Reporting summary

Further information on research design is available in the Nature Portfolio Reporting Summary linked to this article.

## Data availability

Unless otherwise stated, all data supporting the results of this study can be found in the article, supplementary, and source data files. Source data are provided with this paper.

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

## Acknowledgements

A.M. acknowledges support from Alexander von Humboldt Foundation. D.M. acknowledges the support by the start-up funds from DZNE, the grants from the German Research Foundation (SFB 1286/B10), ERC Grant (101078172), and the Human Frontiers Science Organization (RGEC32/2023). Authors would like to acknowledge Dr. Helge Ewers for the support with mass photometry experiments.

## Author contributions

A.M. and R.D. conceived the experiments and designed the project. R.D. supervised the project. R.L developed the theoretical framework. A.M. performed most of the experiments. E.S. performed the microelectrophoresis experiments. K.V.S. performed mass photometry experiments. E.S. and K.V.S contributed equally to this work. C.H. and D. M. expressed and purified the full-length Syn 1 and EGFP-Syn1 proteins and assisted with the phase separation assays and condensate-vesicle interactions. A.M., K.V.S., and E.S. analyzed the data. A.M. and R.D. wrote the paper, with input from the rest of the authors.

## Funding

## Competing interests

The authors declare no competing interests.
