## [Transparent Peer Review file · Nature Communications]

Lipid packing and cholesterol content regulate membrane wetting and remodeling by biomolecular condensates

Corresponding Author: Dr Rumiana Dimova

Version 0:

Reviewer comments:

Reviewer #1

(Remarks to the Author)

The manuscript by Dimova and co-workers describes the effect of lipid membrane organization on the interaction of membranes with biomolecular condensates. Using fluorescence probes and imaging techniques, it was shown that membrane fluidity correlates with the wetting of the membrane by the condensates. Moreover, a clear preference of the glycinin-based condensate to fluid phases in the ternary mixtures was shown. The membrane-protein interaction also provoked membrane remodeling leading to formation of membrane tubes. The dependence of membrane wetting on fluidity was also shown for another type of condensates, which suggests generality of the phenomenon. Overall, this work provides new insights on the aspect of membrane-condensate interaction, which was not sufficiently studied to date. The manuscript is well written and the conclusions are supported by the data. It will be of interest for a broad community of readers. I recommend its publication in Nature Communications after revisions.

Comments:

- 1) Page 8 and Figure 4: it is shown that condensates prefer fluid phases. However, the imaging data reveal some critical difference in the case of gel/fluid vs liquid ordered/liquid disordered phases. In the latter case, the presence of condensates seems to exclude green staining (Figure 4D), which is not the case of gel/fluid phases, where a clear co-localization of the two colors is seen (Figure 4B). This observation may impact the interpretation of data presented in page 8. The authors should present 2D images with sections clearly showing green and red staining in order to see whether there is an impact of the condensate-membrane contact on the membrane (green) staining.
- 2) The paragraph on membrane remodeling is not clearly written and should be improved. The effects of condensates are mixed with effects of salinity, osmolarity and protein adsorption, so it is not clear what is the driving force of observed changes in the membrane morphology. Does this section deal with protein-membrane interactions rather than condensate membrane interactions? The title of this section deals with the protein-membrane interactions, while the title of Figure 5 is focused on the effect of condensates. Moreover, the effects of salinity should be tested without changes in the osmolarity of the solution, for example using a mixture of glucose and sodium chloride, otherwise the two effects are mixed together.
- 3) The authors present the second example of condensate and claim that the phenomenon is universal. I am not sure this is sufficiently justified given that there is a large variety of different condensates and it is not clear how much these observations can be extended to them. Moreover, it would be also important to show how general this observation in terms of lipid composition. The authors used Lo/Ld mixture using DPPC/DOPC + cholesterol. However, the ternary mixture of sphingomyelin/DOPC + cholesterol is more common to study lipid domains and it is closer to natural membranes. Therefore, it would be important to extend the present study to this ternary mixture, showing that the phenomenon is independent of the lipid type.
- 4) Page 18: "Our results using non-tethered 3D condensates reveal that lipid packing alone, in the absence of specific interactions, dictates lipid phase specificity (Figure 4)." This statement is OK, however, in the second example, negatively charged membranes doped with DOPS are studied with condensates based on positively charged peptides. These condensates are tethered due to electrostatic interactions. Therefore, the connection between these two cases should be provided in the discussion.
- 5) Page 4: "The spectral shift between DPPC and DOPC membranes is almost 50 nm, one of the highest reported,". It is not clear whether it is one of the highest reported for Laurdan or in comparison with other solvatochromic dyes. If it is the latter, the authors should be more precise when they do this comparison and provide corresponding references to support this claim.

Reviewer #2

(Remarks to the Author)

In this study, the authors explore how lipid packing influences the wetting/binding affinity of protein condensates to model lipid bilayer membranes. Through the use of hyperspectral imaging and phasor analysis, they quantitatively assessed lipid packing and obtained fluid-elastic parameters with good reliability. The lipid packing was carefully modulated using various lipids, including the addition of cholesterol, and the results are clearly presented and thoroughly analyzed. The data consistently support the main conclusion that protein condensates exhibit weaker binding to more densely packed lipid membranes, which was validated with both glycinin protein and peptide (K10-D10) condensates.

However, the primary concern is if this work represents significant enough scientific and technological advances over the authors' previous studies, particularly in light of their recent publication on related topics (Nat. Commun. 2023, 14, 6081; "Biomolecular condensates modulate membrane lipid packing and hydration"). In that prior work, the binding of glycinin condensates to model membranes was also studied in relation to membrane hydration and lipid packing. Key experimental methodologies, such as hyperspectral imaging and phasor plot analysis, were already reported in this earlier work. While the finding that lipid packing density dictates protein condensate affinity is undoubtedly intriguing, readers may expect more substantial advancements from the present study.

Therefore, I believe that this work would be better suited for publication in Nature Communications if the authors expand its scope by providing additional evidence of innovation beyond their previous contributions and this current work. For example, new experimental approaches could be designed to offer more mechanistic insights into how lipid packing hinders condensate binding and to explore the driving forces behind condensate-membrane interactions. Or, it would be also highly interesting to extend the current analytical system to bio-membranes extracted from cells.

Specific comments & questions

1. While the authors claim the 'universality' of their observations, they only examined two condensate systems (glycinin and K10-D10). To support such a broad conclusion, a wider range of biomolecular condensates must be investigated. For instance, condensates formed from intrinsically disordered proteins (IDPs), which are among the most commonly studied phase-separating proteins, should be tested. Additionally, since glycinin and K10-D10 are highly charged, it would be essential to examine condensates with lower charge densities (or even more hydrophobic) to assess whether the observed trends hold across systems with differing electrostatic properties.

2. Similarly, it is also crucial to vary the condensate systems to better understand the driving forces behind condensate-membrane interactions. For both the highly charged glycinin and K10-D10 condensates, electrostatic interaction is likely the dominant factor. Testing less charged condensates would provide valuable insight into the role of charge in these interactions. Additionally, exploring the effects of varying lipid charges more extensively while controlling lipid packing would further clarify the mechanisms at play. It would also be worthwhile to examine more rigid condensates to see how condensate rigidity influences their binding behavior to membranes.

3. The authors have clearly distinguished specific-tethered condensate binding to membranes from non-specific condensate binding. How would tethered condensate binding be influenced by target membrane lipid packing? A comparison between non-specific condensate interactions, as explored in this study, and specific condensate binding to membranes would be crucial for a deeper understanding of lipid-condensate interactions. NTA-lipids can be easily applied for recruiting 6His-tagged protein condensates for such studies.

4. With specific tethered binding, the binding affinity can also be modulated. For example, the binding affinity between NTA-lipids and 6His-tagged proteins could be controlled by varying coordinated metal ion concentrations. Varying the affinity would provide an opportunity to study the relationship between condensate-membrane binding strength and the influence of lipid packing. This could offer deeper insights into how lipid packing affects condensate behavior depending on the strength of their interaction.

5. The present results should be closely discussed in relation to the previous study on condensate binding and lipid packing (Nat. Commun. 2023, 14, 6081). In that study, condensate binding was found to induce lipid packing, whereas the current findings suggest that lipid packing hinders condensate binding. What accounts for this discrepancy? A thorough examination of these contrasting observations is essential for a comprehensive understanding of the underlying mechanisms.

6. While the use of hyperspectral imaging and phasor analysis is extensively explained (e.g. 2nd & 3rd paragraphs in page), it remains unclear whether these methods have been newly developed or are simply newly applied in this work. Please clarify what, if any, new methodologies have been introduced or developed in this study.

Reviewer #3

(Remarks to the Author)

In their manuscript, Mangiarotti et al demonstrate a clear relationship between lipid membrane fluidity and membrane wetting for 3D condensates without specific membrane tethering. The authors utilize hyperspectral imaging of LAURDAN with phasor analysis to precisely measure the lipid packing/membrane fluidity of giant unilamellar vesicles (GUVs). They also measure the wetting behavior of glycinin or K10/D10 condensates on these GUVs to relate with lipid packing

measurements. The current work builds on the authors' previous work related to how condensates influence membrane fluidity, published in 2023 in Nature Communications (ref 22 of the current manuscript) as well as the corresponding author's experience with the glycerin condensate model published in 2020 in ACS Macro Letters (Chen et al, ref 45 of the current manuscript). This work describes a general mechanism by which lipid packing (and therefore composition) can regulate condensate wetting and possibly also regulate membrane shape generation.

In my opinion, this manuscript is ready for publication after considering the following points:

- Showing that condensate wetting decreases with decreased membrane fluidity (or increased packing) for the Lo phase would strength the overall conclusion that lipid packing and not lipid phase regulates condensate wetting. Figure S1 attempts to do this but does not find a significant trend with the tested lipid compositions. Is there a way to fluidize the Lo other than adding cholesterol? Can temperature be used to fluidize DPPC:Chol Lo vesicles sufficiently to see wetting of these membranes by condensates? Per Fig 1d of Chen 2020 ACS Macro Letters, glycerin condensates can form at higher temperatures.
- Given the resolution limitations of optical microscopy, is there a meaningful minimum detectable intrinsic contact angle? Or resolution limitations in measuring θ_e , θ_c , or θ_i ?
- Can the authors comment on how often membrane tubulation is observed for conditions in Figure 5 and 6?
- The findings in Figure 5 and 6 do not clearly fit the manuscript title or the main findings (Figure 9) of the manuscript. Additional context/clarification should be given in the text to help the reader understand the significance of this data.
- More specific details for GUV deflation should be provided for others to reproduce these results.
- In the second panel of Figure 7A, the GUV membrane (green) appears to wrap the condensate (yellow). Is this representative of the wetting behavior of this condition? Is it still meaningful to apply the same geometric equations shown in Figure 2B to this membrane-wrapped geometry?

Minor edits/typos:

- On page 2, "trough" should be "through".
- On page 4, "HIS" should be "HSI".
- On page 5, the reference to "Figure 3B" is likely meant to be to "Figure 1B".
- On page 9, the reference to "Figure 1 and S1" pertaining to Lo phase data is likely meant to be "Figure 2 and S1".
- On page 11, the reference to "Figure 5D" pertaining to bending rigidity is meant to be "Figure 5E".
- On page 24, in the "Contact angles measurement and geometric factor calculation" section, "Wic and Wic are the respective..." should be "Wic and Wie are the respective..." and " θ_e , θ_e , θ_e " is meant to be " θ_e , θ_c , θ_i ".

Version 1:

Reviewer comments:

Reviewer #1

(Remarks to the Author)

In the revised manuscript, the authors addressed well all my concerns and made new experiments to support their claims. Now I can recommend this manuscript for publication in the present form.

Reviewer #2

(Remarks to the Author)

The authors properly justified the importance of their work and addressed raised questions. In particular, the universality of their observation is better supported now with revision.

Reviewer #3

(Remarks to the Author)

I appreciate the authors' thoughtful responses to my comments and their efforts to address the concerns raised during the review process. After reviewing the revised manuscript and considering their rebuttal, I find that the authors have sufficiently addressed my concerns. I believe the manuscript now meets the standards for publication.

We thank all the reviewers for their comments and suggestions. We revised the manuscript adding new experiments and improving its clarity. All changes and additions are highlighted in blue in the new version of the manuscript. Below you can find the point-by-point response to the comments.

REVIEWER COMMENTS

Reviewer #1:

The manuscript by Dimova and co-workers describes the effect of lipid membrane organization on the interaction of membranes with biomolecular condensates. Using fluorescence probes and imaging techniques, it was shown that membrane fluidity correlates with the wetting of the membrane by the condensates. Moreover, a clear preference of the glycinin-based condensate to fluid phases in the ternary mixtures was shown. The membrane-protein interaction also provoked membrane remodeling leading to formation of membrane tubes. The dependence of membrane wetting on fluidity was also shown for another type of condensates, which suggests generality of the phenomenon. Overall, this work provides new insights on the aspect of membrane-condensate interaction, which was not sufficiently studied to date. The manuscript is well written and the conclusions are supported by the data. It will be of interest for a broad community of readers. I recommend its publication in Nature Communications after revisions.

We thank the reviewer for considering our work interesting and scientifically sound. We have answered the reviewer's comments point by point (see below), and we believe that with the new experiments and clarifications the manuscript has improved and strengthened.

Comments:

1) Page 8 and Figure 4: it is shown that condensates prefer fluid phases. However, the imaging data reveal some critical difference in the case of gel/fluid vs liquid ordered/liquid disordered phases. In the latter case, the presence of condensates seems to exclude green staining (Figure 4D), which is not the case of gel/fluid phases, where a clear co-localization of the two colors is seen (Figure 4B). This observation may impact the interpretation of data presented in page 8. The authors should present 2D images with sections clearly showing green and red staining in order to see whether there is an impact of the condensate-membrane contact on the membrane (green) staining.

We assume that the 3D image projection in Fig 4D led to confusion as it did not show the membrane signal (green) which overlaps with the condensate signal (magenta). Please note that the respective 2D cross sections show this. We have now redistributed and expanded Figure 4 and Figure S2 to include the individual channels showing that there are no discrepancies and better visualizing the phase-specific binding. In both cases, for binary and ternary phase-separated lipid mixtures, the condensates interact only with the liquid-disordered phase. The two individual channels and the provided intensity line profiles demonstrate the co-localization. Below we include the re-organized figures (including the new experiments required in point 3):

Figure 4. Lipid packing determines wetting phase specificity in phase-separated membranes. (A) Sketch illustrating that in presence of gel/fluid (L_d) or liquid-ordered (L_o)/liquid-disordered (L_d) phase-separated GUVs, the condensates bind exclusively to the liquid-disordered (L_d) phase. (B-D) Confocal sections (x,y), line profiles, and 3D projections (x,y,z), of phase-separated GUVs showing that glycinin condensates (magenta) only interact with the fluid or liquid-disordered phase (green), excluding the gel or the liquid-ordered phase. Scale bars: $5\mu\text{m}$. GUVs were labeled with 0.1 mol% ATTO647 N-DOPE and the lipid compositions are: (B) DOPC:DPPC 1:1; (C) DPPC:DOPC:Chol 1:1:1 and (D) DPPC:SM:Chol 1:1:1; cross sections and 3D projections correspond to the same vesicle-condensate couple with the specific membrane composition. The line profiles show that condensates are always interacting with the membrane segments of highest intensity which corresponds to the L_d phase. Dashed lines in the 3D projections are a guide to the eye indicating the vesicle contour. See also Movies S1-S3. All images were taken under the working conditions defined above.

Figure S2. (A) 3D projections of vesicles composed of DOPC:DPPC 1:1, DOPC:DPPC:Chol 1:1:1, and DOPC:SM:Chol 1:1:1 labeled with 0.1 mol% ATTO 647N-DOPE (green). Because ATTO 647N-DOPE preferentially partitions into the liquid-disordered phase, fluorescence is brighter in this phase, while the gel and liquid-ordered phases appear darker. (B) Large field image showing vesicles of the indicated compositions in contact with FITC-labeled glycinin condensates (magenta). In all cases the condensates only interact with the phase presenting lower lipid packing (L_d). (C) Examples of vesicles of the indicated binary and ternary mixtures in contact with glycinin condensates. Individual channels are shown with the corresponding line profiles indicating that condensates only interact with the less packed phases (liquid-disordered). All scale bars: 5 μ m.

2) The paragraph on membrane remodeling is not clearly written and should be improved. The effects of condensates are mixed with effects of salinity, osmolarity and protein adsorption, so it is not clear what is the driving force of observed changes in the membrane morphology. Does this section deal with protein-membrane interactions rather than condensate-membrane interactions? The title of this section deals with the protein-membrane interactions, while the title of Figure 5 is focused on the effect of condensates.

We thank the reviewer for the suggestions. We have renamed and split the sections for clarity. Section 5 is now titled: “**Effect of membrane composition and bending rigidity on membrane remodeling by biomolecular condensates.**” and (new) section 6 is “**Nanotube and double-membrane sheet formation driven by protein adhesion and spontaneous curvature**”. Accordingly, we have modified and improved the text to explain better the observed effects. We have also now modified the sketch in Figure 9 summarizing the main findings:

Figure 9. Sketch summarizing the main findings. (A) The wetting affinity of biomolecular condensates is higher for less densely packed membranes, and can be tuned by changing the lipid chain length, the degree of chain saturation, or the cholesterol content. (B) When in contact with phase-separated membranes, condensates preferentially interact with the less densely packed domains, locally increasing the lipid packing¹. (C) Protein adsorption from homogeneous protein solutions can induce membrane spontaneous curvature, triggering the formation of necklace-like pearls and tubes. When substantial excess area is available, these structures may interconvert into double-membrane sheets (top). Upon interaction with vesicle membranes with excess area, condensates can induce tubulation with tube size depending on the rigidity of the membrane (bottom).

Moreover, the effects of salinity should be tested without changes in the osmolarity of the solution, for example using a mixture of glucose and sodium chloride, otherwise the two effects are mixed together.

Regarding this, we modified the text to clarify that the images shown in Figure 6B were indeed obtained under isotonic conditions (the Figure caption was correct in the previous version). The images shown in Figure 6C,D, and G were taken under isotonic and symmetric conditions (same concentration of NaCl inside and outside the vesicles). We have clarified the use of deflation in the Material and methods section. Note that vesicle deflation, achieved by slightly increasing the concentration/osmolarity of the external solution, only produces an increase in the excess membrane available, but does not determine the observed behaviors, such as the directionality of tubes and curvature generation for example.

3) The authors present the second example of condensate and claim that the phenomenon is universal. I am not sure this is sufficiently justified given that there is a large variety of different condensates and it is not clear how much these observations can be extended to them.

To confirm the universality of our findings, we included two additional systems: PEG/dextran condensates and Synapsin1 condensates. These systems have very different properties, in terms of surface tension, viscosity, hydrophobicity, and surface charge (see new Figures S10 and S11, and Table S1, included below). Despite these differences, they follow the same trend observed for glycinin and K₁₀/D₁₀ coacervates (new Figure 7), strengthening our hypothesis and suggesting that the effects of lipid packing on the condensate-membrane affinity are broadly applicable across various condensate systems. Nevertheless, we toned down the claim in page 20 to "...suggesting this is a universal phenomenon". In addition, we revised the title of section 7 to reflect this.

Figure 7. Influence of lipid packing on condensate wetting is universal across different condensate systems. (A-C) Confocal microscopy images of condensates isolated and in contact with *GUVs* labeled with 0.1 mol% ATTO 647N-DOPE for the indicated membrane compositions. (A) PEG/dextran labeled with 0.5% FITC-dextran (cyan) (B) EGFP-Synapsin I. (C) K_{10}/D_{10} labeled with 0.1 mol% of TAMRA- K_{10} (yellow). (D) Geometric factor Φ (black circles), and intrinsic contact angle θ_e^{in} (cyan circles, right axis), for PEG/dextran condensates in contact with *GUVs* of the indicated compositions. (E) Geometric factor Φ for Syn1 (green diamonds) and K_{10}/D_{10} condensates (yellow circles) in contact with vesicles of the indicated compositions. (F) Intrinsic contact angle θ_e^{in} for Syn1 (green diamonds) and K_{10}/D_{10} condensates (yellow circles) in contact with membranes of the indicated compositions. Individual data points are shown for each membrane composition. The lines indicate the mean value \pm SD. All scale bars: 5 μ m.

Figure S10. Condensates micropolarity measured by ACDAN spectral phasors. (A) Spectral phasor plot of ACDAN in the various condensate systems, with reference data for ACDAN in water and ethanol (EtOH) included. For PEG/dextran condensates, measurements were conducted directly in the bulk dextran-rich phase of the phase-separated system. (B) Left panel: pixel distribution histograms of the data shown in (A). Right panel: center of mass of the distributions plotted in the left panel highlighting differences in micropolarity across the condensate systems. The values for water and EtOH are indicated with dashed lines for reference. (C) Cursor-colored images illustrating variations in dipolar relaxation between the points indicated by the arrow in (A) providing direct visualization of micropolarity differences.

Figure S11. Condensates ζ -potential measured by the microelectrophoresis method² (see Material and Methods for details). (A) Bright-field microscopy images showing the time sequence of the displacement of glycinin condensates in the opposite direction of the externally applied electric field ($E=5$ V/cm). The colored circles are guides to the eye highlighting the trajectory of condensates positions. Scale bar is 20 μm . (B) Drift velocity of condensates with different radius migrating in electric fields of 5 V/cm (glycinin and $K_{10}D_{10}$) and 10-30 V/cm (Syn1) (C) ζ -potentials of the different condensates computed from equation (7). Individual data points are shown, and the mean \pm SD values are indicated on top of each box plot.

Table S1: Summary of the material properties of the tested condensate systems.

Condensate	Viscosity (Pa.s)	Surface tension ($\mu\text{N/m}$)
Glycinin (ref. ¹)	195	15.7
Synapsin 1 (ref. ³)	250	23
PEG/dextran (ref. ⁴ and ⁵)	0.07	8
K_{10}/D_{10} (ref. ⁶ and this work)	0.08	17

Moreover, it would be also important to show how general this observation in terms of lipid composition. The authors used Lo/Ld mixture using DPPC/DOPC + cholesterol. However, the ternary mixture of sphingomyelin/DOPC + cholesterol is more common to study lipid domains and it is closer to natural membranes. Therefore, it would be important to extend the present study to this ternary mixture, showing that the phenomenon is independent of the lipid type.

To ensure that the observed effect was solely due to lipid packing, we aimed to use the same polar headgroup whenever possible, specifically the phosphatidylcholine headgroup. This was the rationale behind choosing the ternary DOPC:DPPC:Chol mixture. However, we took into account the criticism of the reviewer, and have now included measurements with the canonical DOPC:SM:Chol mixture. Our results show that the condensates similarly bind exclusively to the Ld phase, as already shown for DOPC:DPPC:Chol (see Fig. 4 and S2 in the answer to the first comment). This addition clearly strengthens our conclusion that lipid packing, regardless of the headgroup chemistry, directs the specificity of condensate binding to phase-separated membranes.

4) Page 18: “Our results using non-tethered 3D condensates reveal that lipid packing alone, in the absence of specific interactions, dictates lipid phase specificity (Figure 4).” This statement is OK, however, in the second example, negatively charged membranes doped with DOPS are studied with condensates based on positively charged peptides. These condensates are tethered due to electrostatic interactions. Therefore, the connection between these two cases should be provided in the discussion.

We thank the reviewer for the observation, and we have discussed this point more thoroughly. The updated paragraph reads:

“Our results using non-tethered 3D condensates reveal that lipid packing alone, in the absence of specific protein-lipid interactions, dictates condensate specificity for a particular lipid phase (Figure 4). Electrostatics often dominates membrane-condensate interactions, with lipid charges playing a regulatory role in membrane-condensate affinity^{7,8,9}. Here, charged lipids were employed to increase the initial condensate-membrane affinity for Syn1 and K_{10}/D_{10} systems. As shown in Figure S7, membranes composed of DOPC, DLPC, and DPPC, each containing 10% DOPS, exhibit differences in lipid packing density in the order DPPC:DOPS > DLPC:DOPS > DOPC:DOPS. Based on this

trend, one might expect that increased lipid packing, while maintaining the same fraction of charged lipids, would raise the charge density and thereby strengthen the condensate-membrane affinity. However, the opposite effect was observed: membranes with higher packing density exhibited reduced condensate-membrane affinity. This pattern, consistent with observations for neutral membranes, indicates that increased lipid packing weakens affinity even in systems where electrostatics favors the interactions. These findings, validated for two very different systems, Syn1 and K_{10/10}, clearly point to the role of lipid packing as a modulator of condensate-membrane interaction, extending beyond purely electrostatic interactions.”

5) Page 4: “The spectral shift between DPPC and DOPC membranes is almost 50 nm, one of the highest reported,”. It is not clear whether it is one of the highest reported for Laurdan or in comparison with other solvatochromic dyes. If it is the latter, the authors should be more precise when they do this comparison and provide corresponding references to support this claim.

We thank the reviewer for pointing this out, we have corrected it and included proper citations:

“The spectral shift between DPPC and DOPC membranes is ~50 nm, one of the highest reported for membrane solvatochromic dyes^{10, 11}, highlighting LAURDANs sensitivity to subtle changes in membrane packing and hydration^{12, 13, 14}.”

Reviewer #2:

In this study, the authors explore how lipid packing influences the wetting/binding affinity of protein condensates to model lipid bilayer membranes. Through the use of hyperspectral imaging and phasor analysis, they quantitatively assessed lipid packing and obtained fluid-elastic parameters with good reliability. The lipid packing was carefully modulated using various lipids, including the addition of cholesterol, and the results are clearly presented and thoroughly analyzed. The data consistently support the main conclusion that protein condensates exhibit weaker binding to more densely packed lipid membranes, which was validated with both glycinin protein and peptide (K10-D10) condensates. However, the primary concern is if this work represents significant enough scientific and technological advances over the authors' previous studies, particularly in light of their recent publication on related topics (Nat. Commun. 2023, 14, 6081; "Biomolecular condensates modulate membrane lipid packing and hydration"). In that prior work, the binding of glycinin condensates to model membranes was also studied in relation to membrane hydration and lipid packing. Key experimental methodologies, such as hyperspectral imaging and phasor plot analysis, were already reported in this earlier work. While the finding that lipid packing density dictates protein condensate affinity is undoubtedly intriguing, readers may expect more substantial advancements from the present study.

Therefore, I believe that this work would be better suited for publication in Nature Communications if the authors expand its scope by providing additional evidence of innovation beyond their previous contributions and this current work. For example, new experimental approaches could be designed to offer more mechanistic insights into how lipid packing hinders condensate binding and to explore the driving forces behind condensate-membrane interactions. Or, it would be also highly interesting to extend the current analytical system to bio-membranes extracted from cells.

We thank the reviewer for the detailed evaluation of our work and for recognizing its clarity and thoroughness. We appreciate the opportunity to address the reviewer's concern about the "scientific and technological advance" relative to our previous work. While our previous work focused on the effects that protein condensates exert on membranes upon wetting, the current study primarily investigates *why* condensates wet or dewet different membranes. We believe this work offers several key innovations in terms of the scientific conclusions and the methodological integration:

- We constructed a **comprehensive fluidity scale** using LAURDAN hyperspectral imaging and phasor analysis, which we **integrated with fluid-elastic parameters** obtained from the microscopy images. This combination allowed us to draw an impactful conclusion: lipid packing serves as a tunable factor for condensate affinity for the membrane.
- We demonstrate that it is **the degree of lipid packing rather than the phase state** that dictates the **specificity for a particular phase in phase-separated membranes**. This insight was possible **only because we were able to measure and construct the fluidity scale, enabling us to compare the lipid packing** across different binary and ternary mixtures (see e.g. Figure S3). We believe this represents a methodological improvement.
- For the first time, we show that **protein adsorption can drive the formation of nanotubes that interconvert into double-membrane sheets**. Moreover, we demonstrated that this behavior arises from spontaneous curvature generated by protein adsorption, which stabilizes nanotubes at the condensate-membrane interface.
- In response to the suggestions of this reviewer and reviewer 1, we applied our analysis to multiple condensate systems proving that the observed behavior **can be extended across systems with various material and electrical properties**, further strengthening the main hypothesis of the manuscript.
- Finally, we established a **general correlation between the condensate affinity and the membrane fluidity** and we quantified the **tensions of different membrane segments** (see Fig.

7). This new analysis provides a robust mechanistic framework for understanding condensate-membrane interactions.

These advancements are both novel and significant bringing us a step closer to understanding the mechanisms underpinning the membrane-condensate interactions.

The main aim of this work was to test whether lipid packing alone influences the affinity of non-anchored condensates for membranes. As noted by the reviewer, we have provided compelling evidence supporting this hypothesis. Additionally, we have performed new experiments based on the reviewer's suggestions and addressed all comments point-by-point, further enhancing the manuscript clarity and strength. We hope the reviewer finds the revised manuscript suitable for publication in Nature Communications in its current form.

Specific comments & questions

1. While the authors claim the 'universality' of their observations, they only examined two condensate systems (glycinin and K10-D10). To support such a broad conclusion, a wider range of biomolecular condensates must be investigated. For instance, condensates formed from intrinsically disordered proteins (IDPs), which are among the most commonly studied phase-separating proteins, should be tested. Additionally, since glycinin and K10-D10 are highly charged, it would be essential to examine condensates with lower charge densities (or even more hydrophobic) to assess whether the observed trends hold across systems with differing electrostatic properties.

We thank the reviewer for their valuable suggestion. In response we have now tested two additional systems: PEG/dextran condensates and the full-length intrinsically disordered protein Synapsin 1. We observed results consistent with those obtained with glycinin and K₁₀/D₁₀ (see new Figure 7 below), further supporting our hypothesis.

To assess the electrostatic and material properties of the condensate systems tested, we used the solvatochromic dye ACDAN to measure micropolarity. The data reveal that the four condensate systems span a wide range of micropolarities with glycinin and Synapsin1 being the most hydrophobic and PEG/dextran and K10/10 the most hydrophilic (see new Figures S10 and S11 below).

We also evaluated the surface charge of the condensates using microelectrophoresis, a technique recently reported for the accurate measurement of the zeta-potential of condensates². Our measurements show that all the protein/peptide condensate systems exhibit low surface charge. Note that the PEG/dextran condensates are essentially uncharged, given their neutral polymeric nature. For the protein- and the peptide-based condensates, the low zeta-potential values are likely due to the high ionic strength of the buffer (150 mM NaCl for glycinin and Syn1; and 15 mM KCl + 0.5mM MgCl₂ for K₁₀/D₁₀). Note that the zeta-potential value obtained for K₁₀/D₁₀ matches the one reported by another group using the same approach², validating our measurements.

Overall, we have now tested very different condensates systems that differ not only in their material properties (see new Table S1 below) but also in their electrostatic properties, observing a similar trend for all of them, which strengthens our hypothesis for universality. The new pieces of text added to the manuscript include:

“To determine whether the dependence of wetting affinity on membrane lipid packing applies broadly rather than specific to glycinin, we extended our study to other condensate systems with different chemical and material properties. These include: (i) condensates formed by the neutral polymers PEG

and dextran, (ii) condensates formed by the full-length intrinsically disordered protein Synapsin 1 (Syn1), and (iii) condensates formed by two oppositely charged oligopeptides.

Condensates formed by mixtures of PEG and dextran have been extensively studied and are a hallmark of segregative phase-separation^{15, 16}. These condensates exhibit ultralow interfacial tension and low viscosity compared to most protein- or peptide-based condensates⁵ (see summary of material properties in Table S1). Despite the neutral nature of PEG and dextran and their minimal interaction with membranes¹⁵, PEG/dextran condensates can induce extensive membrane remodeling¹⁶. They were the first model system in which condensates have been shown to wet and remodel membranes^{17, 18}. As shown in Figure 7A, when PEG/dextran condensates are brought into contact with vesicles of increasing lipid packing, the condensate/membrane affinity decreases, following a similar trend to that the observed for glycinin (Figure 2).

Next, we tested the interaction of membranes with the full-length protein Syn1. Syn1 is the most abundant synaptic phosphoprotein, and contains a large IDR (a.a. 416-705) that has been shown to be necessary and sufficient for triggering phase separation in vitro^{19, 20}. Syn1 condensates have low affinity for neutral membranes, but their interaction can be significantly enhanced by incorporating negatively charged lipids²¹. Thus, to test how lipid packing affects Syn1 condensate-membrane affinity, it was essential to begin with conditions where the condensate-membrane interaction is robust for membranes with low lipid packing. For this reason, to enhance condensates-membrane interaction, we prepared GUVs made of DOPC, DLPC, and DPPC with 10 mol% DOPS (all forming homogeneous membranes). The phasor plot and fluidity fraction histograms are shown in Figure S7. Inclusion of the charged DOPS increased membranes fluidity (Figure S8) and reduced the fluidity difference between DOPC and DLPC (compare Figures 1C and S7C). Figure 7B shows that when Syn1 condensates are in contact with charged GUVs of increasing lipid packing, the condensate-membrane affinity decreases, further corroborating our findings.”

Figure 7. Influence of lipid packing on condensate wetting is universal across different condensate systems. (A-C) Confocal microscopy images of condensates isolated and in contact with *GUVs* labeled with 0.1 mol% ATTO 647N-DOPE for the indicated membrane compositions. (A) PEG/dextran labeled with 0.5% FITC-dextran (cyan) (B) EGFP-Synapsin I. (C) K_{10}/D_{10} labeled with 0.1 mol% of TAMRA- K_{10} (yellow). (D) Geometric factor Φ (black circles), and intrinsic contact angle θ_e^{in} (cyan circles, right axis), for PEG/dextran condensates in contact with *GUVs* of the indicated compositions. (E) Geometric factor Φ for Syn1 (green diamonds) and K_{10}/D_{10} condensates (yellow circles) in contact with vesicles of the indicated compositions. (F) Intrinsic contact angle θ_e^{in} for Syn1 (green diamonds) and K_{10}/D_{10} condensates (yellow circles) in contact with membranes of the indicated compositions. Individual data points are shown for each membrane composition. The lines indicate the mean value \pm SD. All scale bars: 5 μ m.

Figure S10. Condensates micropolarity measured by ACDAN spectral phasors. (A) Spectral phasor plot of ACDAN in the various condensate systems, with reference data for ACDAN in water and ethanol (EtOH) included. For PEG/dextran condensates, measurements were conducted directly in the bulk dextran-rich phase of the phase-separated system. (B) Left panel: pixel distribution histograms of the data shown in (A). Right panel: center of mass of the distributions plotted in the left panel highlighting differences in micropolarity across the condensate systems. The values for water and EtOH are indicated with dashed lines for reference. (C) Cursor-colored images illustrating variations in dipolar relaxation between the points indicated by the arrow in (A) providing direct visualization of micropolarity differences.

Figure S11. Condensates ζ -potential measured by the microelectrophoresis method² (see Material and Methods for details). (A) Bright-field microscopy images showing the time sequence of the displacement of glycinin condensates in the opposite direction of the externally applied electric field ($E=5$ V/cm). The colored circles are guides to the eye highlighting the trajectory of condensates positions. Scale bar is 20 μm . (B) Drift velocity of condensates with different radius migrating in electric fields of 5 V/cm (glycinin and $K_{10}D_{10}$) and 10-30 V/cm (Syn1) (C) ζ -potentials of the different condensates computed from equation (7). Individual data points are shown, and the mean \pm SD values are indicated on top of each box plot.

2. Similarly, it is also crucial to vary the condensate systems to better understand the driving forces behind condensate-membrane interactions. For both the highly charged glycinin and K10-D10 condensates, electrostatic interaction is likely the dominant factor. Testing less charged condensates would provide valuable insight into the role of charge in these interactions. Additionally, exploring the effects of varying lipid charges more extensively while controlling lipid packing would further clarify the mechanisms at play. It would also be worthwhile to examine more rigid condensates to see how condensate rigidity influences their binding behavior to membranes.

We thank the reviewer for their insightful suggestions.

First, we have demonstrated that none of the condensates tested are highly charged. In fact, they all exhibit low ξ -potential values (see our response to the previous comment). Additionally, the tested condensates vary significantly in their hydrophobicity and material properties, providing a broad range for comparison.

To further address the reviewer's concern, we have now included values of surface tension and viscosity for all condensates tested. PEG/dextran and K_{10}/D_{10} condensates exhibit low viscosities ($\eta = 70\text{--}80$ mPa·s), while glycinin and Synapsin1 condensates are highly viscous, with $\eta = 195$ Pa·s and $\eta = 250$ Pa·s, respectively (summarized in the new Table S1, see below). These values span a wide range, clearly demonstrating that condensate viscosity does not significantly influence wetting affinity. We have now clarified this point in the main text:

“The tested condensates systems exhibit significant variability in material properties including viscosity, surface tension, hydrophobicity, and surface charge (summarized in Table S1 and Figures S10 and S11). ACDAN, is the soluble analog of LAURDAN, which has been previously used to detect changes in the dipolar relaxation in condensates nano-environment using hyperspectral imaging^{1, 22}. By analyzing the spectral phasors of ACDAN, we compared the micropolarity of the different condensates, with pure water and ethanol serving as reference points (Figure S10). The condensates systems span over a wide range of dipolar relaxation values, with glycinin and Syn1 being the most hydrophobic, and PEG/dextran and K_{10}/D_{10} exhibiting a more hydrophilic nano-environment. Additionally, we evaluated the surface charge of the protein/peptide-based condensates by measuring the ζ -potential using microelectrophoresis². We observed that despite that proteins and peptides can be highly charged in homogeneous aqueous solutions (e.g. glycinin ζ -potential $\approx -30\text{mV}$ in water²³), all condensates exhibit low ζ -potential (within a 0-13mV range) under the experimental conditions (see Figure S11). This is likely due to the high ionic strength of the buffer solutions together with the screening that can occur due to protein reorganization during LLPS. Note that the PEG/dextran condensates are essentially neutral and thus this system has not been included in the analysis.”

Table S1: Summary of the material properties of the tested condensate systems.

Condensate	Viscosity (Pa.s)	Surface tension ($\mu\text{N/m}$)
Glycinin (ref. ¹)	195	15.7
Synapsin 1 (ref. ³)	250	23
PEG/dextran (ref. ⁴ and ⁵)	0.07	8
K₁₀/D₁₀ (ref. ⁶ and this work)	0.08	17

Regarding the reviewer's suggestion to explore "more rigid" condensates, we are somewhat puzzled by this request, as condensates are inherently fluid in nature. However, by varying condensate viscosity across orders of magnitude, we have indirectly addressed this aspect, showing that differences in material properties, including "rigidity" inferred from viscosity or interfacial tension, do not significantly impact condensate wetting affinity.

Regarding lipid charges, we confirm that increasing membrane charge enhances condensate affinity for both K₁₀/D₁₀ and Synapsin1, as previously reported^{9, 21}. However, because our primary objective was to investigate the effect of lipid packing on condensate affinity, we deliberately kept the membrane charge content low and fixed to promote a higher initial affinity in the case of the K₁₀/D₁₀ and synapsin1 systems. This rationale is now better explained in the text, and we mention that the effect of lipid charges in the condensate-membrane affinity has been reported^{7, 8}:

"Electrostatics often dominates membrane-condensate interactions, with lipid charges playing a regulatory role in membrane-condensate affinity^{7, 8, 9}. Here, charged lipids were employed to increase the initial condensate-membrane affinity for Syn1 and K₁₀/D₁₀ systems. As shown in Figure S7, membranes composed of DOPC, DLPC, and DPPC, each containing 10% DOPS, exhibit differences in lipid packing density which are ordered according to DPPC:DOPS > DLPC:DOPS > DOPC:DOPS. Based on this trend, one might expect that increased lipid packing, while maintaining the same fraction of charged lipids, would raise the charge density and thereby strengthen the condensate-membrane affinity. However, the opposite effect was observed: membranes with higher packing density exhibited reduced condensate-membrane affinity. This effect, which is consistent with observations for neutral membranes, indicates that increased lipid packing weakens affinity even in systems where electrostatics favors the interactions. These findings, validated for two very different systems, Syn1 and K₁₀/D₁₀, clearly point to the role of lipid packing as a modulator of condensate-membrane interaction, extending beyond purely electrostatic interactions."

3. The authors have clearly distinguished specific-tethered condensate binding to membranes from non-specific condensate binding. How would tethered condensate binding be influenced by target membrane lipid packing? A comparison between non-specific condensate interactions, as explored in this study, and specific condensate binding to membranes would be crucial for a deeper understanding of lipid-condensate interactions. NTA-lipids can be easily applied for recruiting 6His-tagged protein condensates for such studies.

Studies involving NTA-lipids have already been extensively performed, particularly by the group of Jeanne Stachowiak^{24, 25}. In tethered systems, such as those employing NTA-lipids, proteins are "forced" to interact with the membrane through a coordination bond, which, while not covalent, is relatively strong²⁶. This is evidenced by observations where protein condensates that form in bulk will wet membranes only if they contain NTA-lipids, with the affinity increasing proportionally to the concentration of NTA-lipids^{24, 27}. For instance, Stachowiak's group demonstrated that proteins tethered via NTA-lipids can bind to membranes composed of sphingomyelin with 50 mol% cholesterol, forming two-dimensional condensates and remodeling membranes into nanotubes, regardless of the high lipid packing (see e.g. Fig. 4 in ref. ²⁴).

Additionally, the lipid headgroup influences the localization of NTA-lipids, which can partition into either the liquid-disordered or liquid-ordered phase²⁸, allowing targeted interactions with specific phases. This highlights the complexity of tethered systems compared to non-specific interactions. For these reasons, we distinguish specific binding (e.g., NTA-lipid recruitment) from non-specific condensate-membrane interactions, as explored in this study.

Specific binding introduces additional layers of complexity, especially when considering other mechanisms such as GPI-anchored proteins or lipid-specific interactions. While the reviewer rightly highlights that comparing specific and non-specific binding would deepen our understanding of lipid-condensate interactions, this falls beyond the scope of the current study. Nevertheless, it represents an exciting avenue for future research. We have now incorporated this discussion into the main text:

“While in this work we focused on 3D non-anchored condensates, evaluating the impact of lipid packing in systems with specific lipid-protein interactions could provide a deeper insight into the similarities and differences in interaction mechanisms. When 2D condensates form at the membrane surface via protein binding to NTA lipids, the condensate-membrane affinity can be regulated by varying the concentration of NTA lipids^{24, 27}. However, when specific protein-lipid interactions drive membrane wetting, predicting the effect of lipid packing might be challenging, since lipid sorting could arise upon condensate interaction. Moreover, in the case of NTA mediated protein binding, fluorescence quenching by nickel²⁹ complicates the use of fluorescence-based techniques, such as those employed in this work, by affecting the dye lifetime and quantum yield. Alternative systems, such as the specific interaction between the epsin1 N-terminal homology (ENTH) domain and PI(4,5)P2 lipids²⁵, could provide a suitable approach to address this issue. However, exploring the effect of lipid packing in systems with specific protein-lipid interactions is beyond the scope of this work. Nonetheless, it is important to note that NTA-lipids, often used to investigate tethered condensates, are synthetic and not naturally present in biological membranes. This further underscores the value of studying non-tethered or naturally tethered (e.g. via PIP lipids or GPI anchors) condensates to gain insights into physiologically relevant interactions with natural membranes.”

4. With specific tethered binding, the binding affinity can also be modulated. For example, the binding affinity between NTA-lipids and 6His-tagged proteins could be controlled by varying coordinated metal ion concentrations. Varying the affinity would provide an opportunity to study the relationship between condensate-membrane binding strength and the influence of lipid packing. This could offer deeper insights into how lipid packing affects condensate behavior depending on the strength of their interaction.

While it is true that varying the affinity between NTA-lipids and 6His-tagged proteins by adjusting metal ion concentrations could provide useful insights, studying such systems presents some challenges. Specifically, it may be difficult to decouple lipid packing from sorting effects induced by condensate binding. Additionally, the use of nickel and other metals in these systems can act as effective quenchers of fluorescence²⁹, which affects both the intensity and lifetime of the dyes used. These quenching effects would interfere with lipid packing measurements, preventing the quantitative analysis we conducted in this work, and could also complicate the quantification of protein fluorescence at the membrane surface. This would pose difficulties in accurately measuring

the amount of protein bound to the membrane. Nonetheless, we agree that these considerations are important and have now discussed them in the main text as specified in the response to the previous point.

5. The present results should be closely discussed in relation to the previous study on condensate binding and lipid packing (Nat. Commun. 2023, 14, 6081). In that study, condensate binding was found to induce lipid packing, whereas the current findings suggest that lipid packing hinders condensate binding. What accounts for this discrepancy? A thorough examination of these contrasting observations is essential for a comprehensive understanding of the underlying mechanisms.

We thank the reviewer for this important question. To address this, we have expanded the discussion in the manuscript and updated the schematic in Figure 9 (see below) to incorporate all relevant findings (see also our response to comment 2 of Reviewer 1). Importantly, there are no contradictions between the studies. Our previous work demonstrated that condensate adsorption locally increases membrane packing upon interaction, a result corroborated by recent molecular dynamics simulations³⁰. Additionally, FRAP measurements on the membrane in contact with the condensate have shown reduced lipid diffusion compared to bare membranes across various condensate systems^{9, 25}, which reinforces an increased lipid packing at the membrane-condensate interface.

The current study, however, investigates a different question: **why** condensates wet or dewet specific membranes (and not what the changes in the membrane are upon wetting). Our findings indicate that condensates exhibit higher affinity for membranes with low lipid packing. This could be explained by the role of membrane hydration in the interaction process. For condensate wetting to occur, a well-hydrated membrane is required (which may facilitate the dehydration of the interface, a necessary step for binding). Conversely, membranes with higher lipid packing, which are less hydrated, may reduce the interaction potential, leading to lower condensate affinity. Although experimental validation of this precise mechanism remains challenging, our results strongly suggest that lipid packing modulates condensate-membrane interactions via its influence on interface hydration.

The following additional clarification was added to the discussion:

“These results underscore the crucial role of the lipid interface in mediating the interaction. Considering that the water activity at the interface decreases with increasing lipid packing³¹, the dynamics of the interfacial water most likely influences the condensate-membrane interaction. One plausible mechanism is that the interaction between the condensate and the membrane requires dehydration of the interface. In other words, condensates, exhibit a preference for well-hydrated membranes. This would explain why tightly packed membranes, which are already dehydrated, show reduced affinity for condensates compared to loosely packed, highly hydrated membranes. Then, upon interaction, the condensate-membrane affinity drives a localized increase in lipid packing¹.”

A Higher affinity for less packed membranes

B Wetting selectivity for less packed domains

C Morphological transformations and curvature generation

Figure 9. Sketch summarizing the main findings. (A) The wetting affinity of biomolecular condensates is higher for less densely packed membranes, and can be tuned by changing the lipid chain length, the degree of chain saturation, or the cholesterol content. (B) When in contact with phase-separated membranes, condensates preferentially interact with the less densely packed domains, locally increasing the lipid packing¹. (C) Protein adsorption from homogeneous protein solutions can induce membrane spontaneous curvature, triggering the formation of necklace-like pearls and tubes. When substantial excess area is available, these structures may interconvert into double-membrane sheets (top). Upon interaction with vesicle membranes with excess area, condensates can induce tubulation with tube size depending on the rigidity of the membrane (bottom).

6. While the use of hyperspectral imaging and phasor analysis is extensively explained (e.g. 2nd & 3rd paragraphs in page), it remains unclear whether these methods have been newly developed or are simply newly applied in this work. Please clarify what, if any, new methodologies have been introduced or developed in this study.

While hyperspectral imaging (HSI) and phasor analysis are not newly developed techniques, they are not widely used in the community. Here, we apply these methods innovatively to establish a fluidity scale that precisely and sensitively distinguishes differences in membrane chemistry. To help readers understand this framework, we included an explanation and an example in Figure 1, which introduces the fluidity scale used throughout the work. Similarly, the geometric factor, intrinsic contact angle, and affinity contrast definitions in the sketch shown in Figure 2 provide foundation for analysis of the condensate-membrane interactions.

As clarified above, the strength of our work lies not in developing a new methodology but in the novel combination of existing techniques to derive significant insights. The integration of HSI with the fluid-elastic parameters obtained from the microscopy images, has not been employed before. Furthermore, to our knowledge, this study is the first to report the tension of the two membrane segments of a vesicle interacting with a condensate, i.e. be able to compare the tensions of a wetted and a bare membrane segment, a measurement that is hardly accessible through other experimental methods. We have now emphasized all this in the text:

“The approach used in this work, namely combining hyperspectral imaging of LAURDAN to build a fluidity scale and determining fluid-elastic parameters from the microscopy images, allowed us to determine the tensions of the wetted and bare membrane segments for different membrane systems in contact with glycinin condensates (Figure 8C). This information, which is difficult to obtain by other experimental methods, further confirms our previous studies showing that at higher condensate-membrane affinities the lipids at the condensate-membrane interface are pulled together triggering interfacial ruffling when there is enough excess membrane^{1,9}.”

Finally, while the combination of HSI and phasor analysis and fluid-elastic parameters constitutes the essence of our approach, we have employed several other advanced techniques. These include fluctuation spectroscopy (to determine bending rigidity), STED microscopy (to measure nanotube sizes and double-membrane sheets), mass photometry (to measure protein adsorption to the membrane), and microelectrophoresis (to measure condensates z-potential). This multifaceted approach underscores the robustness and depth of our experimental framework.

Reviewer #3:

In their manuscript, Mangiarotti et al demonstrate a clear relationship between lipid membrane fluidity and membrane wetting for 3D condensates without specific membrane tethering. The authors utilize hyperspectral imaging of LAURDAN with phasor analysis to precisely measure the lipid packing/membrane fluidity of giant unilamellar vesicles (GUVs). They also measure the wetting behavior of glycinin or K10/D10 condensates on these GUVs to relate with lipid packing measurements. The current work builds on the authors' previous work related to how condensates influence membrane fluidity, published in 2023 in Nature Communications (ref 22 of the current manuscript) as well as the corresponding author's experience with the glycinin condensate model published in 2020 in ACS Macro Letters (Chen et al, ref 45 of the current manuscript). This work describes a general mechanism by which lipid packing (and therefore composition) can regulate condensate wetting and possibly also regulate membrane shape generation.

In my opinion, this manuscript is ready for publication after considering the following points:

We thank the reviewer for the positive feedback and for the suggestions to improve our work. Below, we have addressed the reviewer's points, and hope the updated version meets the standards for publication.

- Showing that condensate wetting decreases with decreased membrane fluidity (or increased packing) for the L_0 phase would strengthen the overall conclusion that lipid packing and not lipid phase regulates condensate wetting. Figure S1 attempts to do this but does not find a significant trend with the tested lipid compositions. Is there a way to fluidize the L_0 other than adding cholesterol? Can temperature be used to fluidize DPPC:Chol L_0 vesicles sufficiently to see wetting of these membranes by condensates? Per Fig 1d of Chen 2020 ACS Macro Letters, glycinin condensates can form at higher temperatures.

We appreciate the reviewer's suggestion to explore how temperature could be used to fluidize the L_0 phase, and we acknowledge the experimental limitations that prevent us from fully addressing this. While increasing temperature could indeed make the membrane more fluid, as the reviewer mentioned, it also alters the properties of the condensates (such as surface tension and viscosity), complicating the interpretation of any observed changes in wetting. In this case, we would not be able to distinguish whether any increased wetting is due to changes in membrane fluidity or condensate properties.

However, in response to this challenge, we have now included a new ternary mixture, DOPC:SM:Chol (1:1:1), which exhibits a L_0 phase significantly more fluid than the DOPC:DPPC:Chol (1:1:1) or DPPC:Chol (7:3) mixtures (see new Figure S3 below). Despite the increased fluidity of this L_0 phase, we still observe dewetting behavior. This does not rule out the possibility that other condensates might wet the L_0 phase, but it indicates that the condensate-membrane affinity must be substantially higher than that of the systems we evaluated to overcome the inherent resistance of the L_0 or gel phases to wetting.

Additionally, we have performed new experiments with different condensate systems (see new Figure 7 in response to comment 1 of Reviewer 2), showing that PEG/dextran condensates exhibit much higher affinity for the gel phase than the other condensates tested (glycinin, Synapsin1, and K10/D10). We have expanded the discussion accordingly to address this important point.

Updated Figure S3 containing the new membrane composition DOPC:SM:Chol 1:1:1, and panel D highlighting the fluidity of the different L_0 and L_d phases:

Figure S3. (A) Pixel distribution histogram vs fluidity fraction showing an example for DOPC:DPPC 1:1 (gray circles), DOPC:DPPC:Chol 1:1:1 (black circles), and DOPC:SM:Chol (blue circles) vesicles labeled with 0.5 mol% LAURDAN. The fluidity fractions for the compositions in Figures 1, 3 and S1 are included here for comparison. (B) A continuous color scheme (rainbow) is assigned to the fluidity fraction, as shown on top of panel (A), and the images for DOPC:DPPC 1:1, DOPC:DPPC:Chol 1:1:1, and DOPC:SM:Chol without and in contact with condensates are colored accordingly. As the condensates are not labeled, bright-field images merged with the LAURDAN channel are included for reference. (C) Images of vesicles of the indicated compositions colored with the continuous color scheme shown in (A). (D) Zoomed panels of the plot shown in (A), highlighting the differences between the various liquid-ordered (L_o) phases (upper panel), and liquid-disordered (L_d) phases (lower panel) for the binary and ternary mixtures. The maximum position for the single lipid compositions is indicated with gray dashed lines for reference. The fluidity for the liquid-ordered phase increases in the order: DOPC:Chol 7:3 < DOPC:DPPC:Chol 1:1:1 < DOPC:SM:Chol 1:1:1, while for the liquid-disordered phase the fluidity increases according to: DOPC:DPPC:Chol 1:1:1 < DOPC:Chol 7:3 ≤ DOPC:SM:Chol 1:1:1. All scale bars: 5 μ m.

- Given the resolution limitations of optical microscopy, is there a meaningful minimum detectable intrinsic contact angle? Or resolution limitations in measuring θ_e , θ_c , or θ_i ?

This is an excellent point, and we thank the reviewer for raising it. First, it is important to clarify that the determination of the microscopic contact angles θ_e , θ_c , or θ_i is performed with subpixel resolution, exceeding the optical resolution. This is achieved by fitting the contours of the vesicle membrane segments and condensates to circles, as now stated in the Materials and methods section:

“To measure the apparent contact angles from the confocal microscopy images, we first determine the correct projection from z-stacks by aligning the rotational axis of symmetry of the GUV and the condensate. Otherwise, an incorrect projection will lead to a misleading interpretation of the system geometry and incorrect contact angles. Then, by considering that the vesicle, the droplet, and the vesicle-droplet interface correspond to spherical caps, we fit circles to their contours to extract the corresponding angles from geometry, as previously described⁹.”

As indicated above, the apparent contact angles observed in optical microscopy images can be reliably determined when the rotational axis of symmetry of the GUV and condensates is properly aligned (see Supplementary Figure S11 in Mangiarotti et al., 2023⁹).

Second, and more important, while optical microscopy reveals the “membrane kink” defining the apparent angles (Figure 2B), this kink cannot persist at the nanometer scale, as it would imply infinite bending energy. At the nanoscale, the membrane transitions smoothly between the bare vesicle segment and the one in contact with the condensate, as previously considered theoretically^{32, 33} and confirmed experimentally via super-resolution microscopy³⁴. These super-resolution studies demonstrate that the intrinsic contact angle θ_e^{in} can be measured directly at this scale, where the membrane is smoothly curved (Figure 2B sketch). In our work, we derive θ_e^{in} from the apparent contact angles using the relationship $\cos\theta_e^{in} = (\sin\theta_e - \sin\theta_c)/\sin\theta_i$, as now included in the caption of Figure 2. This approach has been validated against the intrinsic contact angles measured directly with super-resolution microscopy, showing excellent agreement³⁴. We have clarified these points in the main text to emphasize the resolution limits and accuracy of our approach:

“It has been demonstrated that there are no significant differences between deriving the intrinsic contact angle from the apparent ones using optical microscopy or directly measuring it using super-resolution microscopy³⁵. It is important to highlight that for correct determination of the apparent contact angles, it is necessary to ensure that the rotational axis of symmetry of the vesicle and the condensate is properly aligned to obtain the right projection, as explained in detail previously⁹.”

- Can the authors comment on how often membrane tubulation is observed for conditions in Figure 5 and 6?

This is an important question, especially considering the inherent heterogeneity of vesicle preparation in terms of initial membrane tension and excess membrane area. In our observations, one-third of the vesicles in the samples displayed sufficient membrane excess area to form tubes. We have added this information in the main text for clarity:

“Under the working conditions used here, we also observed tubulation in approximately one third of the vesicles within a sample.”

- The findings in Figure 5 and 6 do not clearly fit the manuscript title or the main findings (Figure 9) of the manuscript. Additional context/clarification should be given in the text to help the reader understand the significance of this data.

We agree that the section on membrane remodeling could appear disconnected from the main message of the manuscript. To address this, we updated the manuscript title to reflect the observed remodeling processes. Furthermore, we have reorganized and refined the sections for more clarity. Finally, we modified the sketch in Figure 9 to integrate all the main findings, as detailed in our response to comment 2 by Reviewer #1.

Figure 9. Sketch summarizing the main findings. (A) The wetting affinity of biomolecular condensates is higher for less densely packed membranes, and can be tuned by changing the lipid chain length, the degree of chain saturation, or the cholesterol content. (B) When in contact with phase-separated membranes, condensates preferentially interact with the less densely packed domains, locally increasing the lipid packing¹. (C) Protein adsorption from homogeneous protein solutions can induce membrane spontaneous curvature, triggering the formation of necklace-like pearls and tubes. When substantial excess area is available, these structures may interconvert into double-membrane sheets (top). Upon interaction with vesicle membranes with excess area, condensates can induce tubulation with tube size depending on the rigidity of the membrane (bottom).

- More specific details for GUV deflation should be provided for others to reproduce these results.

We have introduced a paragraph on deflation in the materials and methods section:

“When using different solutions for the vesicle growth and condensate formation, the osmolarities were always matched between the suspensions before mixing. In general, to promote condensate interaction, vesicles should possess an excess membrane area to allow deformation. GUVs samples are typically heterogeneous in terms of membrane tension, different vesicles exhibiting varying amounts of excess membrane. This excess area can be increased by vesicle deflation, which can be achieved by slightly increasing the osmolarity of the external solution (e.g. by approximately 5-10% compared to the internal solution) before mixing.”

- In the second panel of Figure 7A, the GUV membrane (green) appears to wrap the condensate (yellow). Is this representative of the wetting behavior of this condition? Is it still meaningful to apply the same geometric equations shown in Figure 2B to this membrane-wrapped geometry?

Thank you for pointing this out. The misleading appearance of the GUV membrane wrapping the condensate in Figure 7A was due to the presence of membrane debris in the vesicle suspension, which occasionally adhere to the condensates. We agree that in such cases, the interfacial properties of the condensate are altered, and can affect the calculation of the geometric factor and have avoided using such images. To avoid any potential confusion, we have now replaced the image in (now) Figure 7C with a more representative one that better reflects the wetting behavior under these conditions:

Figure 7. Influence of lipid packing on condensate wetting is universal across different condensate systems. (A-C) Confocal microscopy images of condensates isolated and in contact with *GUVs* labeled with 0.1 mol% ATTO 647N-DOPE for the indicated membrane compositions. (A) PEG/dextran labeled with 0.5% FITC-dextran (cyan) (B) EGFP-Synapsin I. (C) K_{10}/D_{10} labeled with 0.1 mol% of TAMRA- K_{10} (yellow). (D) Geometric factor Φ (black circles), and intrinsic contact angle θ_e^{in} (cyan circles, right axis), for PEG/dextran condensates in contact with *GUVs* of the indicated compositions. (E) Geometric factor Φ for Syn1 (green diamonds) and K_{10}/D_{10} condensates (yellow circles) in contact with vesicles of the indicated compositions. (F) Intrinsic contact angle θ_e^{in} for Syn1 (green diamonds) and K_{10}/D_{10} condensates (yellow circles) in contact with membranes of the indicated

compositions. Individual data points are shown for each membrane composition. The lines indicate the mean value \pm SD. All scale bars: 5 μ m.

Minor edits/typos:

- On page 2, “trough” should be “through”.
- On page 4, “HIS” should be “HSI”.
- On page 5, the reference to “Figure 3B” is likely meant to be to “Figure 1B”.
- On page 9, the reference to “Figure 1 and S1” pertaining to Lo phase data is likely meant to be “Figure 2 and S1”.
- On page 11, the reference to “Figure 5D” pertaining to bending rigidity is meant to be “Figure 5E”.
- On page 24, in the “Contact angles measurement and geometric factor calculation” section, “Wic and Wic are the respective...” should be “Wic and Wie are the respective...” and “ $\theta_e, \theta_e, \theta_e$ ” is meant to be “ $\theta_e, \theta_c, \theta_i$ ”.

Corrected.

References

1. Mangiarotti A, Siri M, Tam NW, Zhao Z, Malacrida L, Dimova R. Biomolecular condensates modulate membrane lipid packing and hydration. *Nature Communications* **14**, 6081 (2023).
2. van Haren MHI, Visser BS, Spruijt E. Probing the surface charge of condensates using microelectrophoresis. *Nature Communications* **15**, 3564 (2024).
3. Wang H, *et al.* Live-Cell Quantification Reveals Viscoelastic Regulation of Synapsin Condensates by α -Synuclein. *bioRxiv*, 2024.2007.2028.605529 (2024).
4. Zhao Z, Satarifard V, Lipowsky R, Dimova R. Membrane nanotubes transform into double-membrane sheets at condensate droplets. *Proceedings of the National Academy of Sciences* **121**, e2321579121 (2024).
5. Wang H, Kelley FM, Milovanovic D, Schuster BS, Shi Z. Surface tension and viscosity of protein condensates quantified by micropipette aspiration. *Biophysical Reports* **1**, 100011 (2021).
6. Fisher RS, Elbaum-Garfinkle S. Tunable multiphase dynamics of arginine and lysine liquid condensates. *Nature Communications* **11**, 4628 (2020).
7. Lu T, *et al.* Endocytosis of Coacervates into Liposomes. *Journal of the American Chemical Society* **144**, 13451-13455 (2022).
8. Lu T, Hu X, van Haren MHI, Spruijt E, Huck WTS. Structure-Property Relationships Governing Membrane-Penetrating Behaviour of Complex Coacervates. *Small*, 2303138 (2023).
9. Mangiarotti A, Chen N, Zhao Z, Lipowsky R, Dimova R. Wetting and complex remodeling of membranes by biomolecular condensates. *Nature Communications* **14**, 2809 (2023).
10. Ragaller F, *et al.* Dissecting the mechanisms of environment sensitivity of smart probes for quantitative assessment of membrane properties. *Open Biology* **12**, 220175 (2022).

11. Klymchenko AS. Solvatochromic and Fluorogenic Dyes as Environment-Sensitive Probes: Design and Biological Applications. *Accounts of Chemical Research* **50**, 366-375 (2017).
12. Parasassi T, De Stasio G, Ravagnan G, Rusch RM, Gratton E. Quantitation of lipid phases in phospholipid vesicles by the generalized polarization of Laurdan fluorescence. *Biophysical Journal* **60**, 179-189 (1991).
13. Parasassi T, De Stasio G, d'Ubaldo A, Gratton E. Phase fluctuation in phospholipid membranes revealed by Laurdan fluorescence. *Biophysical Journal* **57**, 1179-1186 (1990).
14. Gunther G, Malacrida L, Jameson DM, Gratton E, Sánchez SA. LAURDAN since Weber: The Quest for Visualizing Membrane Heterogeneity. *Accounts of Chemical Research* **54**, 976-987 (2021).
15. Dimova R, Lipowsky R. Giant Vesicles Exposed to Aqueous Two-Phase Systems: Membrane Wetting, Budding Processes, and Spontaneous Tubulation. *Advanced Materials Interfaces* **4**, 1600451 (2017).
16. Mangiarotti A, Dimova R. Biomolecular Condensates in Contact with Membranes. *Ann Rev Biophys*, (2024).
17. Li Y, Lipowsky R, Dimova R. Transition from complete to partial wetting within membrane compartments. *Journal of the American Chemical Society* **130**, 12252-12253 (2008).
18. Li Y, Kusumaatmaja H, Lipowsky R, Dimova R. Wetting-Induced Budding of Vesicles in Contact with Several Aqueous Phases. *The Journal of Physical Chemistry B* **116**, 1819-1823 (2012).
19. Milovanovic D, Wu Y, Bian X, De Camilli P. A liquid phase of synapsin and lipid vesicles. *Science* **361**, 604-607 (2018).
20. Hoffmann C, et al. Synapsin condensation controls synaptic vesicle sequestering and dynamics. *Nature Communications* **14**, 6730 (2023).
21. Hoffmann C, et al. Electric Potential at the Interface of Membraneless Organelles Gauged by Graphene. *Nano Letters* **23**, 10796-10801 (2023).
22. Ambroggio EE, Costa Navarro GS, Pérez Socas LB, Bagatolli LA, Gamarnik AV. Dengue and Zika virus capsid proteins bind to membranes and self-assemble into liquid droplets with nucleic acids. *Journal of Biological Chemistry* **297**, 101059 (2021).
23. Liu C, et al. Aggregation kinetics and ζ -potential of soy protein during fractionation. *Food Research International* **44**, 1392-1400 (2011).
24. Yuan F, et al. Membrane bending by protein phase separation. *Proceedings of the National Academy of Sciences* **118**, e2017435118 (2021).
25. Lee Y, et al. Transmembrane coupling of liquid-like protein condensates. *Nature Communications* **14**, 8015 (2023).
26. Conti M, Falini G, Samorì B. How Strong Is the Coordination Bond between a Histidine Tag and Ni – Nitriolotriacetate? An Experiment of Mechanochemistry on Single Molecules. *Angewandte Chemie International Edition* **39**, 215-218 (2000).

27. Kang CY, Chang Y, Zieske K. Lipid Membrane Topographies Are Regulators for the Spatial Distribution of Liquid Protein Condensates. *Nano Letters* **24**, 4330-4335 (2024).
28. Wang H-Y, *et al.* Coupling of protein condensates to ordered lipid domains determines functional membrane organization. *Science Advances* **9**, eadf6205 (2023).
29. Pramanik S, Steinkühler J, Dimova R, Spatz J, Lipowsky R. Binding of His-tagged fluorophores to lipid bilayers of giant vesicles. *Soft Matter* **18**, 6372-6383 (2022).
30. Mondal S, Cui Q. Coacervation-Induced Remodeling of Nanovesicles. *The Journal of Physical Chemistry Letters* **14**, 4532-4540 (2023).
31. Socas LBP, Ambroggio EE. Linking surface tension to water polarization with a new hypothesis: The Ling-Damodaran Isotherm. *Colloids and Surfaces B: Biointerfaces* **230**, 113515 (2023).
32. Kusumaatmaja H, Li Y, Dimova R, Lipowsky R. Intrinsic Contact Angle of Aqueous Phases at Membranes and Vesicles. *Physical Review Letters* **103**, 238103 (2009).
33. Lipowsky R. Remodeling of Biomembranes and Vesicles by Adhesion of Condensate Droplets. *Membranes* **13**, 223 (2023).
34. Zhao Z, Roy D, Steinkühler J, Robinson T, Lipowsky R, Dimova R. Super-resolution imaging of highly curved membrane structures in giant vesicles encapsulating molecular condensates. *Advanced Materials* **34**, 2106633 (2021).
35. Zhao Z, Roy D, Steinkühler J, Robinson T, Lipowsky R, Dimova R. Super-Resolution Imaging of Highly Curved Membrane Structures in Giant Vesicles Encapsulating Molecular Condensates. *Advanced Materials* **34**, 2106633 (2022).